# Evaluating World Models with LLMs for Decision Making

## Abstract

World model emerges as a key module in decision making, where MuZero and Dreamer achieve remarkable successes in complex tasks. Recent work leverages Large Language Models (LLMs) as general world simulators to simulate the dynamics of the world due to their generalizability. LLMs also serve as the world model for deliberative reasoning in Reasoning via Planning (RAP) and Tree of Thought (ToT). However, the world model is either evaluated as a general world simulator, or as a functional module of the agent, i.e., predicting the transitions to assist the planning. In this work, we propose a comprehensive evaluation of the world models with LLMs from the decision making perspective. Specifically, we leverage the **31** diverse environments from (Wang et al., 2023; 2024) and curate the rule-based policy of each environment for the diverse evaluation. Then, we design three main tasks, i.e., **policy verification**, **action proposal**, and **policy planning**, where the world model is used for decision making solely. Finally, we conduct the comprehensive evaluation of the advanced LLMs, i.e., GPT-4o and GPT-4o-mini, on the environments for the three main tasks under various settings. The key observations include: i) GPT-4o significantly outperforms GPT-4o-mini on the three main tasks, especially for the tasks which requires the domain knowledge, e.g., scientific tasks, ii) the performance of the world models with LLMs depends predominantly on their performance in key steps, while the total number of steps required for task completion is not a reliable indicator of task difficulty, and iii) the combination of different functionalities of the world model for decision making will brings unstability of the performance and partially obscures the performance gap between stronger and weaker models, e.g., GPT-4o and GPT-4o-mini.

## 1 Introduction

The remarkable achievements of MuZero (Schrittwieser et al., 2020) and Dreamer (Hafner et al., 2019; 2021; 2023) have established world models (Ha & Schmidhuber, 2018) as a fundamental module in decision-making systems. World models serve as learned simulators that encode rich representations of environment dynamics, enabling agents to predict future states based on their actions. By learning to predict how the world evolves in response to actions, these models enable several key capabilities. i) Generalization to novel tasks: World models have demonstrated impressive transfer learning abilities (Byravan et al., 2020), allowing agents to adapt to previously unseen scenarios by leveraging their learned understanding of world dynamics. This generalization capacity is particularly valuable in robotics and control applications where agents must handle diverse situations (Robey et al., 2021; Young et al., 2023). ii) Efficient planning: The predictive capabilities of world models enable the sophisticated planning algorithms (Sekar et al., 2020; Hamrick et al., 2021; Schrittwieser et al., 2020). By simulating possible futures, agents can evaluate different action sequences and select optimal strategies without requiring actual interaction with the environment. This "imagination" or "mental simulation" capability dramatically improves sample efficiency and safety in decision-making. iii) Offline learning: World models have proven especially valuable in offline reinforcement learning settings (Schrittwieser et al., 2021; Yu et al., 2020; 2021), where agents must learn from pre-collected datasets without direct environment interaction. The ability to learn accurate dynamics models from historical data has opened new possibilities for training agents in scenarios where online interaction is impractical or costly. Recent advances have expanded the scope of world models beyond traditional reinforcement learning applications. Systems like Genie (Bruce et al., 2024) and Vista (Gao et al., 2024)

demonstrate how world models can serve as general-purpose simulators that users can directly interact with. These developments suggest a future where world models might serve as foundational building blocks for artificial general intelligence, providing systems with interactive understanding of how the world works.

Large Language Models (LLMs) achieve remarkable success in enormous natural language tasks in the past five years (Brown et al., 2020; OpenAI, 2023). Several recent works leverage LLMs as the general world models to provide the environment knowledge for various complex tasks, e.g., math and reasoning. With the fine-tuning over pre-collected data from the environments, the LLMs can predict the action sequences across different tasks over environments while maintaining the capabilities on other domains (Xiang et al., 2023). LLMs also serve as the world model explicitly in Reasoning via Planning (RAP) (Hao et al., 2023) and Reason for Future, Act for Now (RAFA) (Liu et al., 2023), where the LLMs predict the next states based on the actions executed at current states, e.g., the states of blocks in the BlocksWorld (Valmeekam et al., 2023), which is used to assist the planning methods. LLMs serve as the world model implicitly in the widely-used Tree of Thoughts (ToT) (Yao et al., 2023), as well as Graph of Thoughts (GoT) (Besta et al., 2024), where the LLMs need to predict the states and evaluate the thoughts to help the selection of the thoughts to advance the reasoning. Recent work also consider LLMs as world simulators (Wang et al., 2024; Xie et al., 2024), where they evaluate the performance of LLMs on the prediction of next states and the game progress, demonstrating the potentials of LLMs as general world models.

Most of the previous works evaluate the world models with LLMs either as general world simulators (Wang et al., 2024), or as additional modules of the agents to make decisions (Liu et al., 2023). The comprehensive evaluation of world models from a decision-making perspective has been largely overlooked. This evaluation is crucial for two main reasons. First, general world simulators need to estimate the state transition from any state $s \in S$, while in many environments, only a small portion of the state space will be visited ($S_{\text{visit}} << S$) when computing the optimal policy. For example, AlphaZero finds the super-human policy (Silver et al., 2018) by only exploring a small proportion (less than 1%) of the state space. Therefore, we argue that world models with LLMs should be evaluated in a more decision-oriented perspective, i.e., focusing more on the states relevant to the task at hand ($S_{\text{visit}}$). Second, the influence of the world models is usually coupled with the actors who choose the actions, i.e., if the actor cannot pick the correct actions, the task cannot be completed even when the world model is accurate. The coupling of the actors and the world models brings additional difficulties to understand the world models, brings the obstacles for researchers to build better world models for decision making.

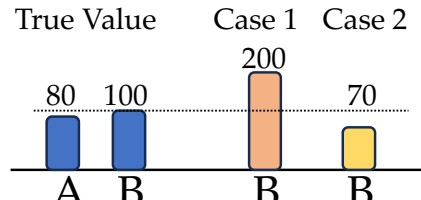

Figure 1: Picking the action with higher value. The true value of A and B are 80 and 100, respectively, therefore, B is the correct action. Case 1's prediction 200 is worse than in Case 2's prediction 70 (compared to the true value 80). However, in Case 1 the most valuable action is B, which is also the case in the true value setting. This shows that more accurate predictions (Case 2) do not always lead to correct decisions (Case 1).

Therefore, we address these issues with three key observations about world models for decision making:
**Observation 1: Prediction is important, but not that important.** An illustrative example is displayed in Figure 1, which indicates that more accurate predictions do not lead to correct decisions.[1] This phenomenon is also observed in other decision making scenarios, e.g., financial trading (Sun et al., 2023). The success of MuZero Unplugged (Schrittwieser et al., 2021) also demonstrate that we can learn good policies from inaccurate world models which are trained only with limited data. This motivates us that the evaluation of the world models for decision making should focus on the predictions which relevant to the desired policy, rather than as general world simulators. Besides, the decision making usually involves multiple steps and the errors of the one-step predictions are accumulated when the number of steps increases. Therefore, the accuracy of the one-step predictions is not adequate for the evaluation of the world model for decision making and novel tasks should be proposed.
**Observation 2: Selecting potential actions should be an important feature for world models.** Most of the previous works in world model focus on next state and reward prediction, and the action se-

---

[1]The issue in Figure 1 can be elicited by various methods, e.g., rank prediction. This example is just to illustrate the discrepancy between prediction and decision, motivating us to reconsider the evaluation of the world model for decision making.

lection is usually completed by the actors, i.e., the model trained for generate a single action for executing. We argue that with more knowledge about the world, the world model may make a better selection of the potential actions. Besides, selecting a set of potential actions, e.g., 10 potential actions, may significantly reduce the difficulties of the tasks and improve the performance when combing with planning. World models can also be viewed as game engines (Valevski et al., 2024), which have to provide potential actions to guide fresh players to complete tasks, e.g., Red Dead Redemption 2 (Tan et al., 2024). Therefore, action proposal should be considered for evaluation, which can be easily implemented for the world models with LLMs.

**Observation 3: Planning with world models can find the policies solely.** With the prediction of the next states and the action proposal, we can leverage planning methods or search methods to find the policies. It is observed that most state-of-the-art methods for complex decision-making tasks, e.g., Chess or Go, is based on the planning with an accurate simulator (Silver et al., 2018; Monroe & Chalmers, 2024) or the world model (Schrittwieser et al., 2020). Most works introduce the critic (i.e., the value function) to evaluate the actions immediately for efficient planning (Schrittwieser et al., 2020; Hao et al., 2023). We note that the critic is not necessary for finding policies and may also influence the performance. Therefore, we argue that only the planning with the next state prediction and the action proposal is necessary when incorporating the world model in decision making. This focused approach allows for better isolation and evaluation of the world model's performance.

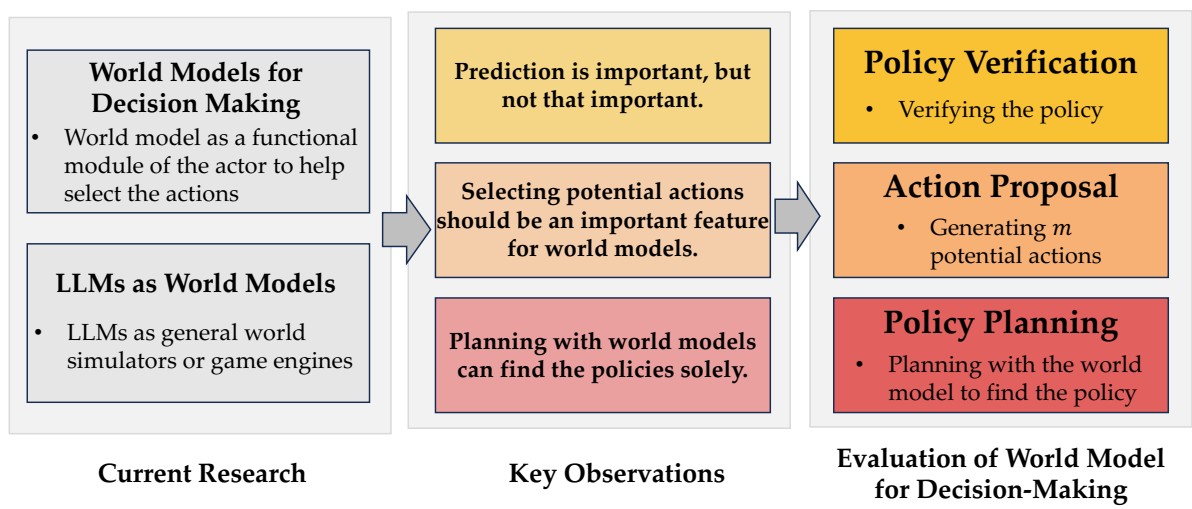

Figure 2: Evaluation of World Model with LLM for Decision Making.

Based on the above three key observations, we propose a comprehensive evaluation of world models with LLMs for decision making. Specifically, we leverage **31 diverse environments** from (Wang et al., 2023; 2024) with different tasks varying from daily tasks, e.g., washing clothes, to scientific tasks, e.g., forging keys, and curate the rule-based policy for each environment for the evaluation. Then, we design three main tasks: i) **policy verification**: verifying whether the policy can complete the task, ii) **action proposal**: proposing the top-$K$ actions that can potentially complete the task, and iii) **policy planning**: finding the policy solely with the combination of the different functionalities, i.e., policy verification and action proposal. Finally, we conduct the comprehensive evaluation of the advanced LLMs, i.e., GPT-4o and GPT-4o-mini, on the environments for the three tasks under various settings. The key observations include: i) GPT-4o significantly outperforms GPT-4o-mini on the three main tasks, especially for the tasks which requires the domain knowledge, e.g., scientific tasks, ii) the performance of the world models with LLMs depends predominantly on their performance in key steps, while the total number of steps required for task completion is not a reliable indicator of task difficulty, and iii) the combination of different functionalities of the world model for decision making will brings unstability of the performance and partially obscures the performance gap between stronger and weaker models, e.g., GPT-4o and GPT-4o-mini.

## 2  Related Work

**World Models in Decision Making.**  World models are actively explored by researchers to further improve the agent's performance and the sample efficiency (Ha & Schmidhuber, 2018; Janner et al., 2019; Hafner et al., 2019; Schrittwieser et al., 2020). Dreamer (Hafner et al., 2019) is a practical model-based reinforcement learning algorithm that introduces the belief over states as a part of the input to the model-free DRL algorithm used. Trajectory Transformer (Janner et al., 2021) trains the transformer to prediction the next state and action as a sequence modeling problem for continuous robot control. MuZero (Schrittwieser et al., 2020) is a remarkable success of model-based RL, which learns the world model and conduct the planning in the latent space. MuZero achieves the superior performances over other model-based and model-free RL methods. The world models trained in these methods are problem-specific and cannot be generalized to other problems, which motivates researchers to seek to more generalizable world models, e.g., LLMs as world models. The world model with LLM in (Xiang et al., 2023) is trained to gain the environment knowledge, while maintaining other capabilities of the LLMs. Dynalang (Lin et al., 2024) proposes the multimodal world model, which unifies the videos and texts for the future prediction in decision making.

**LLMs as World Simulators.**  World simulators are developed to model the dynamics of the world (Bruce et al., 2024). LLMs serve as the world simulator due to their generalizability across tasks. Specifically, The LLMs (i.e., GPT-3.5 and GPT-4) is evaluated to predict the state transitions, the game progress and scores with the given object, action, and score rules, where these rules are demonstrated to be crucial to the world model predictions (Wang et al., 2024). The world models with LLMs in (Xie et al., 2024) need to additionally identify the valid actions. We move a step further to ask the world model to propose the potential actions to complete the tasks (**Observation 2**). Both methods mainly focus on the prediction of the state, which may be not suitable for the evaluation of the world model for decision making (**Observation 1**).

**World Models in LLMs.**  The concept of world model also be explored in the deliberation reasoning of LLMs. Specifically, Reasoning via Planning (RAP) (Hao et al., 2023) leverage the planning methods (e.g., Monte Carlo Tree Search (MCTS)) with the world model with LLMs for plan generation and math reasoning, where LLMs need to predict the next state and the reward to guide the search. Tree of Thought (ToT) (Yao et al., 2023) implicitly leverage the LLMs as the world model to predict the next state and the reward for the search over different thoughts. Reason for future, act for now (RAFA) (Liu et al., 2023) combine the planning and reflection with the world model for complex reasoning tasks. However, these methods do not focus on the evaluation of the world models, and several interdependent modules are coupled with each other for completing the task (**Observation 3**).

## 3  Preliminaries

**Markov Decision Process (MDP).**  A decision making problem is usually represented as a Markov decision process (MDP) (Sutton & Barto, 2018), which is defined by the tuple $(S, A, T, R, \gamma)$, where $S$ is the state space, $A$ is the action space, $T : S \times A \to S$ is the transition dynamics, which specifies the next state $s'$ given the current state $s$ and action $a$, $R : S \times A \to \mathbb{R}$ is the reward function, which specifies the agent's reward given the current state $s$ and action $a$, and $\gamma$ is the discount factor. The agent's policy is defined by $\pi_\theta : \mathcal{S} \times \mathcal{A} \to [0, 1]$, parameterized by $\theta$, which takes the state $s$ as the input and outputs the action $a$ to be executed. The objective of the agent is to learn an optimal policy $\pi^* := \arg\max_\pi \mathbb{E}_\pi \left[ \sum_{t=0}^{\infty} \gamma^t r_t | s_0 \right]$ is the expected return and $s_0$ is the initial state.

**Large Language Models (LLMs).**  Large Language models (LLMs) learn from text data using unsupervised learning. LLMs optimize the joint probabilities of variable-length symbol sequences as the product of conditional probabilities by $P(x) = \prod_{i=1}^{n} P(s_i | s_1, ..., s_{i-1})$, where $(s_1, s_2, ..., s_n)$ is the variable-length sequence of symbols. With the billions of parameters and extensive training data, the vast amounts of common knowledge encoded in LLMs lead to the remarkable generalization across various NLP tasks with simple prompting and in-context learning, without the need for task-specific fine-tuning (Touvron et al., 2023; OpenAI, 2023). Given the generalizability, LLMs present themselves as a promising foundation for constructing comprehensive world models.

**World Models.** The world model $\Omega$ is introduced to predict the dynamics of the environment, thus supporting the decision making process. Specifically, the world model is trained or prompted to predict the next state $s'$, the reward $r$, and the terminal function $d$, given the current state $s$ and action $a$. The world model can be one or multiple neural networks specially trained on the environments for the three prediction tasks (Hafner et al., 2019; Schrittwieser et al., 2020), which cannot generalize across different environments. Recent work leverage LLMs to build the general world models, where the prompting (Xie et al., 2024), in-context learning (Wang et al., 2024), and even fine-tuning methods (Xiang et al., 2023; Lin et al., 2024) are used to transform the LLMs to the world models.

## 4 World Models with LLMs for Decision Making

In this section, we introduce the world model with LLM for decision making. Specifically, we will introduce the next state prediction, the reward and terminal prediction. Then, we will introduce how the world model will be used to complete the considered three main tasks, i.e., policy verification, action proposal, and policy planning. We provide the relationship between the three main tasks and the two kinds of predictions in Figure 3 for better understanding of the rationale behind the three tasks.

The world model considered in this work mainly follows the design in (Wang et al., 2024), where the representation of the states includes the objects in the environments and their properties. The prompts to the LLM, e.g., GPT-4o, also include the object rules, the action rules, and the score rules, which provides the necessary knowledge of the environments for the LLM to make accurate predictions. For the **next state prediction**, we ask the LLM to predict the state changes, i.e., the change of the objects' properties, which is demonstrated to be efficient for the prediction (Wang et al., 2024). With the predicted state changes, we can recover the full state for further predictions. For the **reward/terminal prediction**, the LLM needs to predict three features: i) gameScore: the reward received from the environment, ii) gameOver: whether the task is

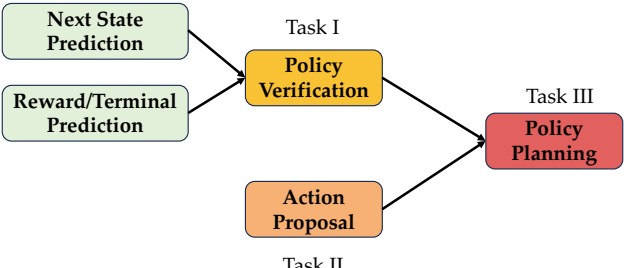

Figure 3: The next state prediction and reward/terminal prediction are considered in (Wang et al., 2024). With the two predictions, we can complete **policy verification**. We introduce the **action proposal** of the world model and the **policy planning** can be completed with the policy verification and action proposal.

terminated, and iii) gameWon: whether the task is successfully completed or not. For the rules used for the prediction, we refer to (Wang et al., 2024) for more details and the code to generate the prompts is also provided in Appendix C.1 for completeness.[2]

**Policy Verification.** Motivated by **Observation 1**, we propose policy verification, one of the most straightforward task to evaluate the world models in terms of the multi-step predictions. The basic idea of policy verification is given an action sequence, the world model need to prediction whether the sequence can complete the task or not. The process for the policy verification is displayed in Algorithm 1. Specifically, given the environment `env`, the action sequence $\boldsymbol{a}$ with length $N$ to verify and the proportion of the action sequence to verify $\rho$, we will run the game for the first $\rho \cdot N$ steps (Line 4 of Algorithm 1), and leverage the world model to continue the last $(1 - \rho)N$ steps (Line 5 of Algorithm 1). The returned results $r_N, d_N$ will be compared with the true results from the environment to evaluate the performance of the world model.

**Action Proposal.** The action proposal is a novel task for world model, based on **Observation 2**. Basically, we will ask the world model to recommend top-$K$ actions that can potentially complete the task. Specifically, we follow the representation of the state in the next state prediction, with the additional information: i) the examples of actions, and ii) the previous actions. The previous actions can help the LLMs to understand the game progress. The code to LLM for the action proposal is displayed in Appendix C.2.

---

[2]Due to the space constraints and the extreme length of the prompts, we cannot provide a complete example in the paper. We will open-source all the codes for readers to replicate our results.

One key issue for the action proposal is that the action generated by the world model may not be valid for the game at the current state. Therefore, given the predicted action $a'$ and the set of possible actions to be executed at the current state $A'$, we leverage the text-embedding model (OpenAI, 2022) to query the most similar actions with the cosine similarity, i.e., $a^* = \arg\max\{\texttt{emb}(a', a), \forall a \in A'\}$.

---

**Algorithm 1:** Policy Verification

---

1 Given the `env`, the action sequence $\boldsymbol{a}$ to verify with $N = \texttt{len}(\boldsymbol{a})$, $\rho$ the proportion of $\boldsymbol{a}$ to verify, the world model $\Omega$
2 $s_0 = \texttt{env}()$;
3 **for** $t \in \{1, 2, ..., N-1\}$ **do**
4    **if** $t < \rho \cdot N$ **then** $s_{t+1}, r_t, d_t = \texttt{env}(a_t)$;
5    **else** $s_{t+1}, r_t, d_t = \Omega(s_t, a_t)$;
6 **return** $r_N, d_N$

---

**Algorithm 2:** Policy Planning

---

1 Given the `env`, the action sequence $\boldsymbol{a}$ with $N = \texttt{len}(\boldsymbol{a})$, $\rho$ the proportion of $\boldsymbol{a}$ for planning, the world model $\Omega$, the planning sequence $\boldsymbol{a}' = []$
2 **for** $t \in \{1, 2, ..., \rho \cdot N\}$ **do**
3    $s_{t+1}, r_t = \texttt{env}(a_t), \boldsymbol{a}'.\texttt{append}(a_t)$
4 **for** $t \in \{\rho \cdot N, \cdots, (2-\rho)N\}$ **do**
5    $a_t = \Omega(s_t), s_{t+1}, r_t, d_t = \Omega(s_t, a_t), \boldsymbol{a}'.\texttt{append}(a_t)$;
6    **if** $d_t$ **then break**;
7 **return** $\boldsymbol{a}'$

---

**Policy Planning.** The policy planning task is motivated by **Observation 3**, which combines the policy verification and the action proposal (as displayed in Figure 3). The process of policy planning is displayed in Algorithm 2. Specifically, we execute the actions in the given $\boldsymbol{a}$ on the environment for $\rho N$ steps (Line 3 in Algorithm 2) and then plan for $2(1-\rho)N$ steps with the world model (Line 5 in Algorithm 2), where both the action to execute and the state transitions are generated by the world model. The returned action sequence $\boldsymbol{a}'$ will be evaluated in the environment to verify the correctness. We note that only top-1 action is generated in Algorithm 2 for illustration. When more actions are generated, we need to enumerate all possible outcomes or leverage advanced search methods, which will be tackled in future work.

## 5 Environments

**Tasks.** We leverage the **31** diverse environments from (Wang et al., 2023)[3] with different tasks varying from daily tasks, e.g., washing clothes, to scientific tasks, e.g., forging keys. This task suite is more related to the real physical world, including the physical objects, e.g., bulb and bathtub, and the iterations with these physical objects, i.e., turn on the hot tap to improve the temperature of the water in the bathtub. Compared with other widely used environments, such as the grid world, e.g., BabyAI (Chevalier-Boisvert et al., 2019) and the web environments, e.g., MiniWob++ (Shi et al., 2017), this task suite is more relevant to the common knowledge encoded in the LLMs. A full list of the descriptions and the taxonomy of the environments can be found in Appendix B.1.

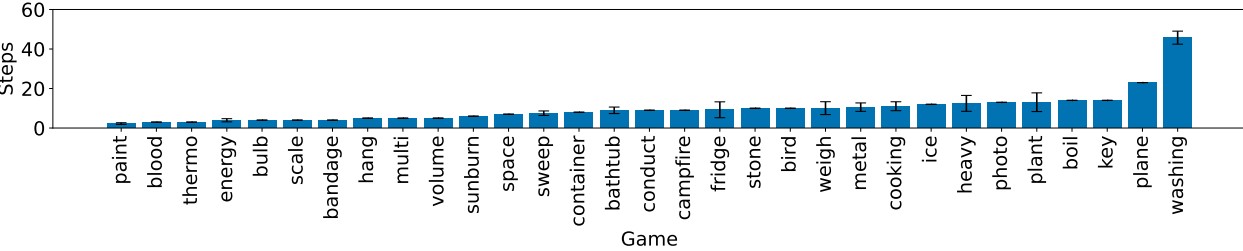

Figure 4: Number of steps to complete the tasks

**Rule-based Policies.** There are various randomness in the environments, including the specific tasks, e.g., the target color can be "orange", "purple", "green", "black" in the paint task. However, for each task, only a single playthrough is provided in (Wang et al., 2024), which is not enough for a comprehensive evaluation

---

[3]We note that there are 32 environments in (Wang et al., 2023) and the dish-washing environment is used as the example for the world model, which is excluded for fair evaluation.

of the world model for decision making. Therefore, we curate the rule-based policy for each environment and verify the correctness for 200 runs. The scripts for the rule-based policies are provided in Appendix B.2, which can help readers to understand the process to complete the tasks, as well as the complexities of tasks. We provide the statics of the number of steps to complete the tasks for 200 runs in Figure 4.[4]

# 6 Evaluations

In this section, we present the comprehensive evaluation of the world model for decision making over the diverse 31 environments on the three main tasks under various setttings. We use the advanced LLM models, i.e., GPT-4o and GPT-4o-mini [5], as the backbone LLM for the world model. We set the temperature of the LLM to be 0 to reduce the variance of the generation and all results are averaged over 30 runs due to the randomness of the environments.

## 6.1 Task I: Policy Verification

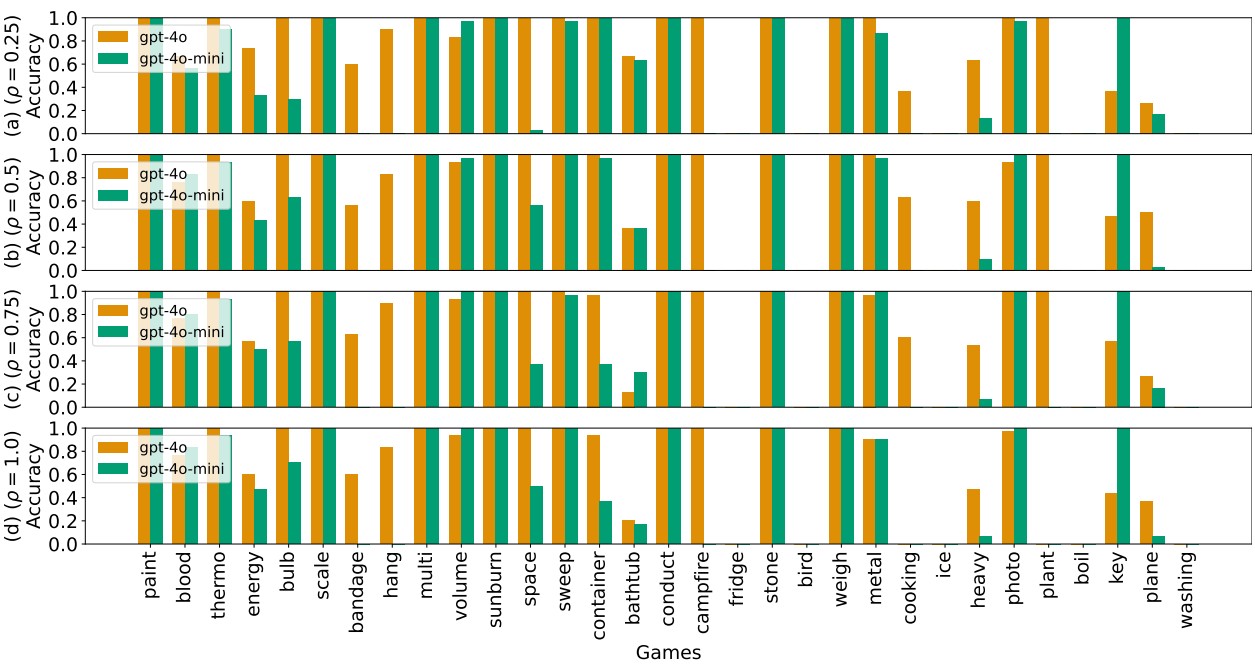

Figure 5: The accuracy of the world model to verify the correct policies

**Evaluation Protocol.** Given the action sequence $a$ generated by the rule-based policy, we leverage the world model to verify the last $\rho$ proportion of the policy, where $\rho \in \{0.25, 0.5, 0.75, 1.0\}$. We note that when $\rho = 1.0$, the world model will verify the full action sequence with only the initial observation of the environment. We say the verification of the policy is correct if all three features, i.e., gameScore, gameOver, and gameWon, are correct. We note that only correct policies are verified, as there are enormous wrong policies for an environment, which is useless for decision making. Furthermore, there would also be other action sequences to finish the tasks, where we cannot enumerate all policies to complete the tasks.

**Evaluation Results.** The policy verification results are displayed in Figure 5.[6] We observe that GPT-4o outperform GPT-4o-mini in most tasks and especially on the tasks which requires the domain knowledge, e.g., bandage, hang, and campfire. We also observe that with more steps of the verified policies, the performance gap between GPT-4o and GPT-4o-mini is increase. With larger proportion of the action sequences to verify,

---

[4]The names on the figure may differs from (Wang et al., 2023) for visualization, and please refer to Table 1 for correspondence.
[5]Due to the limited budget, we do not take Claude and Gemini into consideration.
[6]Note that the result of the policy verification is either 0 or 1, so the error bar is not plotted.

i.e., $\rho$ increase, the accuracy of the verification is decreased, which indicates that the accumulation of the errors in the world model, either on the next state prediction or the reward prediction, will influence the performance of the world model. This observation is consistent to the fact that the LLM may not perform well in long-term decision making tasks. We also observe that more steps to complete the tasks do not necessarily lead to the worse performance, which indicates that the domains of the tasks play a more important role for the policy verification, i.e., for the tasks where the LLM has enough domain knowledge, e.g., conduct, stone, weigh and photo (Wang et al., 2024), the task would be easy even when the number of steps is large. We also provide the accuracy of the three prediction tasks separately in Appendix D, and we found that both GPT-4o and GPT-4o-mini performs worse for predicting gameScore, while performs much better for predicting gameOver and gameWon. This indicates that the value prediction is more difficult for LLMs, which is consistent the observations from other works. During the experiments, both models frequently returned empty dictionaries, suggesting they sometimes failed to properly follow the given instructions.

### 6.2 Task II: Action Proposal

**Evaluation Protocol.** The action proposal requires the world model to generate the top-$K$ potential actions to complete the tasks, where $K \in \{1, 2, 3, 5, 10\}$. Specifically, given the action sequence $\boldsymbol{a}$ generated by the rule-based policies, we will let the world model to generate the potential actions with the states along with the path of $\boldsymbol{a}$ to complete the task. We say the action proposal is correct if the actions in $\boldsymbol{a}$ in the generated actions by the world model. The results of the accuracy are averaged over the steps over the action sequence and 30 runs of each environment. We also note that the action sequence $\boldsymbol{a}$ generated by the rule-based policy is not the only sequence to complete the task and we cannot enumerate all possible actions which can lead to the completion of the task. We note that the number of available actions in the environments is usually larger than 500, which brings difficulties to the traditional RL methods for training and indicate the necessity for the world model to generate the potential actions to facilitate the learning.

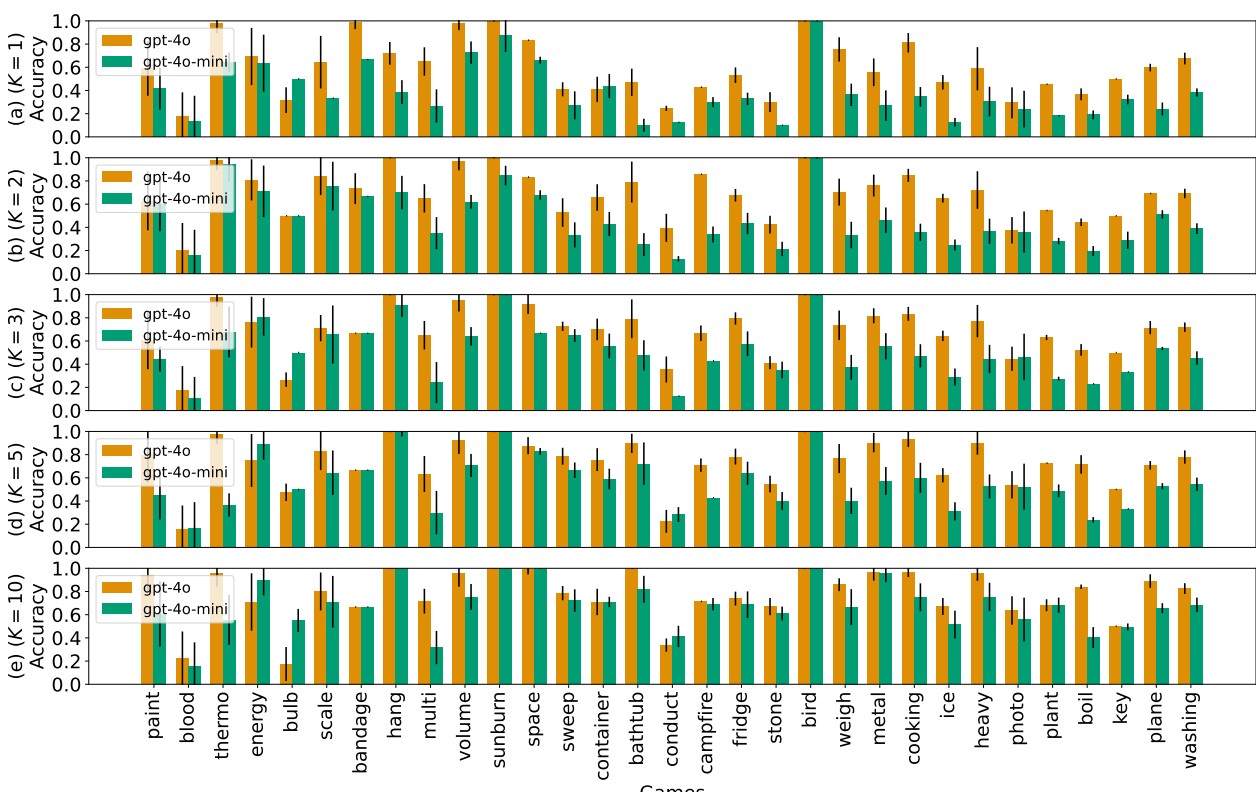

Figure 6: The accuracy of the world model to generate the potential actions

**Evaluation Results.** The action proposal results are displayed in Figure 6. Overall, GPT-4o consistently outperforms GPT-4o-mini across different tasks and different values of $K$. With the increase of the number of steps to complete the tasks, where more analysis of the previous actions is needed to understand the game progress, GPT-4o maintains the better accuracy, while GPT-4o-mini shows a substantial drop of the accuracy. The performance gap between the two models is generally increased when the number of steps to complete the tasks increase. When $K = 10$, the accuracy of the action proposal for GPT-4o is very high in most tasks. With approximately 800 possible actions available at each time step, the results demonstrate that GPT-4o effectively identifies and selects relevant actions while filtering out irrelevant ones. This capability shows promising potential for successful task completion. Furthermore, we still observe that both models obtain lower values in the tasks requiring the domain knowledge, i.e., blood and conduct, which is consistent to the observation in (Wang et al., 2024) that LLMs, e.g., GPT-4, is more likely to make errors when scientific knowledge is needed. We also provide the step accuracy of the action proposal in Appendix E to illustrate the prediction of the relevant actions along with the steps. We observe that there are some key steps that has extremely low accuracies, which indicates that the critical steps significantly influences the difficulties of the tasks, rather than the number of steps to complete the tasks. We also observe that both GPT-4o and GPT-4o-mini can generate wrong actions even when the action rules are given, especially for the environments where the domain scientific knowledge is needed, e.g., 'space-walk'.

### 6.3 Task III: Policy Planning

**Evaluation Protocol.** The policy planning is based on the policy verification and the action proposal, as showed in Algorithm 2. Similar to the policy verification, we let $\rho \in \{0.25, 0.5, 0.75, 1.0\}$ to vary the number of steps for the planning. We only consider the case with $K = 1$, i.e., the world model only generates the top-1 action with the given states. Finally, we evaluate the planned policy $\boldsymbol{a}'$ in the environment to verify the correctness. We note that when $K = 1$, no advanced search method is needed, while when $K > 1$, we cannot enumerate all possible outcomes for larger steps, e.g., 10. Besides, a critic is also needed to choose among the outcomes for verifying in the environments. Therefore, we only consider the case with $K = 1$ and leave the case $K > 1$ into future work.

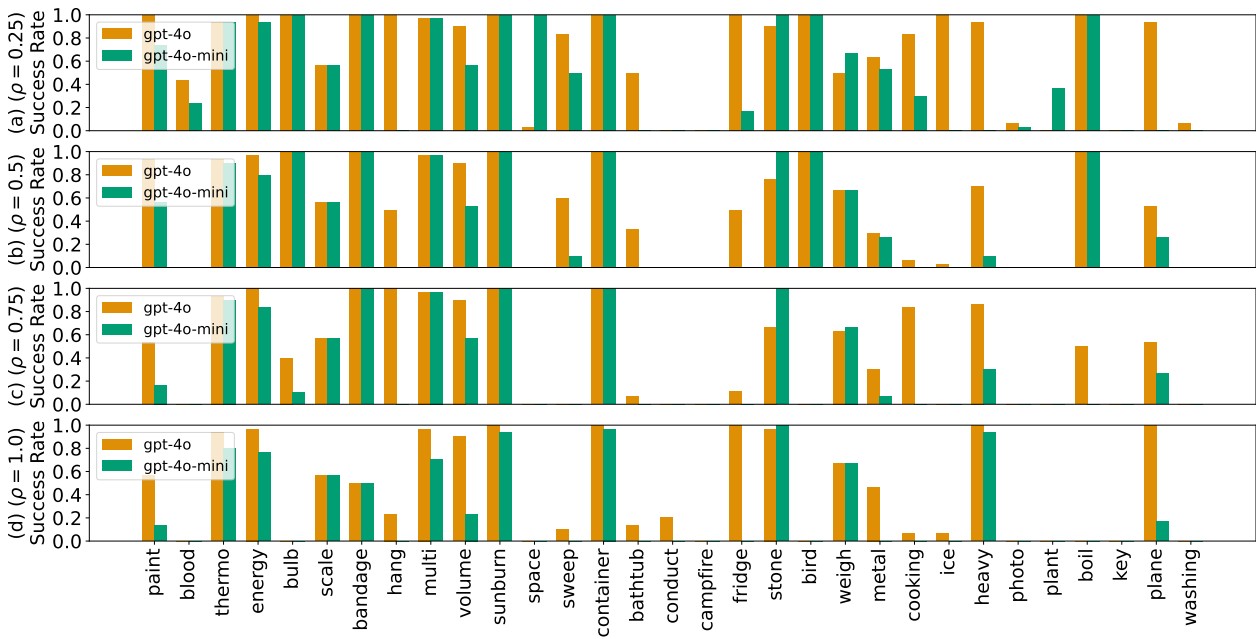

Figure 7: The success rate of the world model to complete the tasks

**Evaluation Results.** The policy planning results are displayed in Figure 7, where GPT-4o and GPT-4o-mini achieve comparable performance for the tasks with less steps and smaller values of $\rho$, e.g., 0.25 and GPT-4o generally achieve better results in tasks with more steps. When the value of $\rho$ increases, the performance

is generally decreasing. With the coupling of the policy verification and action proposal, we observe more unstabilities of the performances of models over tasks and settings. This indicates the necessity of decoupling the functionalities of the world model for the evaluation. Similar to the model-based RL (Schrittwieser et al., 2020), where introducing the world model may bring the training unstabilities, we need to be carefully to apply the world models with LLMs to the decision making due to the inherent complexity. During the experiments, we also observe the format errors of the outputs from both models, which may interrupt the running of the experiments. The frequency of these failures depends on the environments, where the 'hang-painting', 'space-walk', and 'make-campfire' are the three environments we experiences most of the failures. Therefore, with the interaction of the different functionalities of the world model, the system is more unstable due to the unexpected in LLMs' outputs.

## 7  Conclusions

World model is a key module for decision making and recent works leverage LLMs as the general world models. However, the evaluation of the world models with LLMs for decision making is far from satisfactory. In this work, we propose three main tasks, i.e., policy verification, action proposal, and policy planning, to evaluate the performance of the world model. We conduct the comprehensive evaluation of advanced LLMs, i.e., GPT-4o and GPT-4o-mini, on **31** diverse environments over the three main tasks under various settings. The key observations include: i) GPT-4o significantly outperforms GPT-4o-mini on the three main tasks, especially for the tasks which requires the domain knowledge, e.g., scientific tasks, ii) the performance of the world models with LLMs depends predominantly on their performance in key steps, while the total number of steps required for task completion is not a reliable indicator of task difficulty, and iii) the combination of different functionalities of the world model for decision making will brings unstability of the performance and partially obscures the performance gap between stronger and weaker models, e.g., GPT-4o and GPT-4o-mini.

**Limitations and Future Work.** There are several limitations of this work. i) For the policy planning task, we only consider the case with $K = 1$, i.e., the world models only predict 1 potential action. We will tackle the cases with $K > 1$ in future work by introducing the advanced searching methods, e.g., DFS. ii) The tasks considered in this work is relatively straightforward, other complex tasks for utilizing world models for decision making will considered in future work, e.g., training the actor to select the actions by only interacting with the world models and planning with the safety constraints. Solving these complex tasks requires more sophisticated combinations of different functionalities of the world models. iii) The number of environments considered in this work is still limited and more diverse environments will be considered in future work, including the web environments (Zhou et al., 2024), the board games (Li et al., 2023), and the street maps of cities (Vafa et al., 2024). By evaluating world models with LLMs across these comprehensive environments and tasks, we believe world models will become fundamental in guiding decision-making processes, particularly in areas of generalization, safety, and ethical considerations. iv) We only consider the world models with different prompts for different environments. There are several methods which can be incorporated to improve the world models, including the in-context learning (Agarwal et al., 2024), retrieval-augmented generation (Lewis et al., 2020), and fine-tuning (Hu et al., 2022). We will include these advanced methods into the evaluate of the world model in the future work.

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

# A  Frequently Asked Questions

## A.1  More Justifications of the Proposed Tasks

In this section, we will provide a mode detailed justification of the three tasks proposed in this work.

**Policy Verification.**  Policy verification can be viewed as a generalization of the next state prediction. Instead of focusing on the accuracy of the one-step prediction about the next states and the reward/terminal prediction, which is considered in most previous works, policy verification may accumulate the multi-step predictions and judging whether the given policy can complete the task or not. This task is more relevant to the world model for decision making, as if the world model can verify any given policy correctly, with the enough number of sampling of the policy, i.e., action sequences, we can complete the task in the end.

**Action Proposal.**  As observing in the ToT (Yao et al., 2023), generating useful thoughts is critical which can significantly improve the performance. However, with multiple thoughts generated, we have to select one to executed. We can test these thoughts in the environments, however, this is not always doable. Therefore, building a world model is the straightforward way to do this. On the other hand, action proposal is necessary for the world model as the game engine to guide the fresh players to complete the game. With increasing the number of recommend actions, the difficulty of the action proposal is decreased. However, this task is not considered in the previous work for the world model. We believe that this task should be an important task to evaluate the world model for decision making.

**Policy Planning.**  Policy planning is a combination of policy verification and action proposal. Conceptually, if the world model performs well on both tasks, we can obtain the policy with world model only and no actor is needed. This can help us to understand the world model through decoupling the world model with any other modules. Besides, this planning task is consistent with the System 2 thinking, i.e., with more time for the planning, the world model may find better policies.

## A.2  The Objective of This Paper

The primary objective of this paper is proposing the new evaluation tasks for the evaluation of the world models with LLMs for decision making.

- Evaluating world model for decision making is difficult, given that the decision making tasks usually involves multiple steps of the predictions. Therefore, the one-step prediction tasks considered in most previous works is not suitable.

- Instead of treating the world models as world simulators or supporting modules for the actors, we identify that the world model can solve the tasks solely with the combination of the policy verification and action proposal. Therefore, the world model should be researched with the same importance.

- World models have traditionally been evaluated through a top-down approach, where complex systems are constructed to complete tasks, constraining analysis to high-level observations. By examining fundamental capabilities like policy verification and action proposal, we propose a bottom-up evaluation approach that enables more systematic and granular assessment of world models.

## A.3  Selection of Backbone LLMs

- We need to evaluate the three novel tasks over 31 environments, and each with 30 runs, we roughly use 3000 dollars for all the experiments for GPT-4o and GPT-4o-mini. Due to the limited budget, we cannot afford to test on Claude and Gemini.

- For the open-sourced models, we test on some open-sourced models, e.g., Qwen 7B model, and find that current open-sourced models still cannot generate the responses with correct formats, i.e., JSON. This brings difficulties for the evaluation.

### A.4  Comparison with Decision Transformer and Trajectory Transformer

Decision transformer (DT) (Chen et al., 2021) trains the transformer to predict the action conditional on the experiences and the target reward or the goal. Language DT (LDT) (Gontier et al., 2023) extends DT to consider the text-based games and include the state prediction in the training as an auxiliary tasks. However, during the inference, i.e., decision making, the model still generates the action directly, which is not based on the world model because the state prediction is only used for training and not for generating actions.

Trajectory Transformer (TT) (Janner et al., 2021) also consider the decision making problem as a sequence modeling problem, where the transformer is trained to predict the state, the action and the reward. Compared with DT, TT is more related to the world model, where the state and reward prediction is used to generating actions and the search methods, e.g., beam search, is used. However, only the continuous robot control is considered in TT (Janner et al., 2021) and the trained TT model is domain and problem specific, which cannot generalize to other problems.

Recently, LLMs provide a promising way to build the general world model and the world model with LLMs emerge as a novel research field. However, most of these work focus on the single-step prediction and a comprehensive evaluate is needed. This work is inspired by TT and extends the insights from TT to the world model with LLMs for text-based games. Specifically, we consider the policy verification, the action proposal and the policy planning tasks, where the TT combines these tasks to generate the actions and only investigate the performance for the decision. Instead of only considering the performance of the decision makings, our three tasks provide a more bottom-up analysis of the world model for decision making.

# B  Environments

## B.1  Introduction of Tasks

| Environments | Task Description |
| --- | --- |
| mix-paint (paint) | Your task is to use chemistry to create black paint. |
| blood-type (blood) | Your task is to give a correct type of blood to the patient. |
| thermometer (thermo) | Your task is to figure out the temperature of the water in the pot. |
| clean-energy (energy) | Your task is to change all fossil-fuel power stations to use renewable energy while keeping the same capacity. |
| lit-lightbulb (bulb) | Your task is to lit the light bulb. |
| scale-weigh (scale) | Your task is to figure out the weight of the apple. |
| use-bandage (bandage) | Your task is to put bandages on any cuts. |
| hang-painting (hang) | Your task is to hang the picture of a girl (ID: 11) on the back wall (ID: 5). |
| multimeter (multi) | Your task is to figure out the resistance of the resistor 0. |
| volume (volume) | Your task is to figure out the volume of the green box. |
| sunburn (sunburn) | It is a summer noon. The sky is clear. Your task is to take a ball from the beach and put it in the box in the house. Protect yourself from sunburn! |
| space-walk (space) | Your task is to conduct a space walk. |
| sweep-floor (sweep) | Your task is to clean the garbage on the ground to the garbage can. |
| volume-container (container) | Your task is to figure out the volume of the glass. |
| bath-tub-water-temperature (bathtub) | Your task is to make the temperature of the water in the bath tub to 35 - 40 Celsius degree by adding water from the taps. When you are done, take the action "bath". |
| conductivity (conduct) | Your task is to figure out if the fork is conductive or not. If the fork is conductive, put it in the red box. Otherwise, put it in the black box. |
| make-campfire (campfire) | Your task is to make a fire in the fire pit. |
| refrigerate-food (fridge) | Your task is to prevent the foods from spoiling. |
| volume-stone (stone) | Your task is to figure out the volume of the stone. |
| bird-life-cycle (bird) | Your task is to hatch the egg and raise the baby bird. |
| balance-scale-weigh (weigh) | Your task is to figure out the weight of the cube. Use the answer action to give your answer. |
| metal-detector (metal) | Your task is to find the buried metal case on the beach. You win the game by putting the metal case in your inventory. |
| cooking (cooking) | Your task is to prepare a meal following the instructions of the cook book. |
| make-ice-cubes (ice) | Your task is to make ice cubes. |
| balance-scale-heaviest (heavy) | Your task is to put all heaviest cubes into the box. |
| take-photo (photo) | Your task is to take a nice picture of orange (ID: 4), using a camera with shutter speed of 1/2, aperture of 16, and iso of 1600. |
| plant-tree (plant) | Your task is to plant the tree and water it. |
| boil-water (boil) | Your task is to boil water. |
| forge-key (key) | Your task is to forge a key to open the door. |
| inclined-plane (plane) | Here are two inclined planes with the same angle. Your task is figure out which of the two inclined planes has the most friction. Focus on the inclined plane with the most friction after your experiment. |
| wash-clothes (washing) | Your task is to wash the dirty clothes and dry them. |

Table 1: Environments (Wang et al., 2024)

There are **32** environment in (Wang et al., 2023) and dish-washing is selected as the example in the prompt, which is excluded for fair evaluation. Specifically, the environments can be categorized into two domains:

- **Daily-life tasks**, including use-bandage, hang-painting, sunburn, sweep-floor, bath-tub-water-temperature, make-campfire, refrigerate-food, cooking, take-photo, plant-tree, boil-water, and wash-clothes. For these tasks, the world model need to have the common knowledge about the procedure of completing these tasks, e.g.. first collecting the dirty clothes, then put them into the washing machine, then use the detergent and turn on the washing machine for the wash-clothes task.

- **Scientific tasks**, including mix-paint, blood-type, thermometer, clean-energy, lit-lightbulb, scale-weigh, multimeter, volume, space-walk, volume-container, conductivity, volume-stone, bird-life-cycle, balance-scale-weigh, metal-detector, make-ice-cubes, forge-key and inclined-plan. These tasks requires the scientific knowledge to complete the tasks, e.g., the world model need to know that the friction may decrease the speed of a block sliding down of the plane for the inclined-plane. Then, the world model need to generate build a micro-simulation to compare the frictions of the two planes.

## B.2 Code for Demo Actions Generation

Only one playthrough of the game is provided in (Wang et al., 2024), which is not enough due to the randomness in the environments. Therefore, we curate the rule-based policy for each environment. Note that due to the randomness of the environments, e.g., the target color in the mix-paint task, the generated action sequences are different for different instantances of the same environment.

Code Sample 1: mix-paint

```python
def get_demo_actions(self):
    target_color = self.useful_info[0][0]
    paint_names = self.useful_info[1]

    color_dict = {
        # "red": (1, 0, 0),
        "orange": ["red", "yellow"],
        # "yellow": (0, 1, 0),
        "green": ["yellow", "blue"],
        # "blue": (0, 0, 1),
        "purple": ["red", "blue"],
        "black": ["red", "yellow", "blue"],
    }
    paint_names_to_idx = {}
    for paint_idx, paint_name in enumerate(paint_names):
        paint_names_to_idx[paint_name] = paint_idx
    color_mix_plan = color_dict[target_color]

    demo_actions = []
    to_idx = -1
    for color_idx, color_mix in enumerate(color_mix_plan):
        if color_idx == 0:
            to_idx = paint_names_to_idx[color_mix]
            continue
        demo_actions.append(
            "pour {} paint (ID: {}) in cup {} (ID: {})".format(
                color_mix,
                2 * (paint_names_to_idx[color_mix] + 1) + 1,
                to_idx,
                2 * (to_idx + 1),
            )
        )
    demo_actions.append("mix cup {} (ID: {})".format(to_idx, 2 * (to_idx + 1)))

    return demo_actions
```

Code Sample 2: blood-type

```python
def get_demo_actions(self):
    useful_info = self.useful_info[1]

    return [
        "give Type {} {} blood (ID: 3) to patient (ID: 2)".format(
            useful_info[0], useful_info[1]
        ),
        "take Type {} {} blood (ID: 3)".format(useful_info[0], useful_info[1]),
        "give Type {} {} blood (ID: 3) to patient (ID: 2)".format(
            useful_info[0], useful_info[1]
        ),
    ]
```

Code Sample 3: thermometer

```python
def get_demo_actions(self):
    demo_actions = [
        "take thermometer (ID: 4)",
        "use thermometer (ID: 4) on water (ID: 3)",
        "answer {} Celsius degree".format(self.water_temperature),
    ]
    return demo_actions
```

Code Sample 4: clean-energy

```python
def get_demo_actions(self):
    demo_actions = []
    change_station = {
        "sun": "solar farm",
        "water": "hydroelectric power station",
        "wind": "wind farm",
    }
    for region in self.regions:
        demo_actions.append(
            "change {} to {}".format(
                region.name, change_station[region.properties["resource"]]
            )
        )

    return demo_actions
```

Code Sample 5: lit-lightbulb

```python
def get_demo_actions(self):
    return [
        "connect light bulb (ID: 2) terminal1 to red wire (ID: 3) terminal1",
        "connect red wire (ID: 3) terminal2 to battery (ID: 6) anode",
        "connect battery (ID: 6) cathode to black wire (ID: 4) terminal1",
        "connect black wire (ID: 4) terminal2 to light bulb (ID: 2) terminal2",
    ]
```

Code Sample 6: scale-weigh

```python
def get_demo_actions(self):
    demo_actions = [
        "take {}".format(self.useful_info[0].name),
        "put {} on {}".format(self.useful_info[0].name, self.useful_info[1].name),
        "look",
        "answer {}g".format(self.target_weight),
    ]

    return demo_actions
```

Code Sample 7: use-bandage

```python
def get_demo_actions(self):
    demo_actions = [
        "open bandage box (ID: 8)",
        "look",
        "take bandage (ID: 9)",
    ]
    return demo_actions + [
        "put bandage (ID: 9) on {} (ID: 3)".format(self.useful_info[0])
    ]
```

Code Sample 8: hang-painting

```python
def get_demo_actions(self):
    demo_actions = [
        "take nail (ID: 7)",
        "take hammer (ID: 6)",
        "hammer nail (ID: 7) on {} with hammer (ID: 6)".format(
            self.target_wall.name
        ),
        "take {}".format(self.target_picture.name),
        "hang {} on nail (ID: 7)".format(self.target_picture.name),
    ]

    return demo_actions
```

Code Sample 9: multimeter

```python
def get_demo_actions(self):
    demo_actions = [
        "set multimeter (ID: 2) to resistance mode",
        "connect multimeter (ID: 2) terminal1 to resistor {} (ID: 3) terminal1".format(
            self.target_resistor_id
        ),
        "connect multimeter (ID: 2) terminal2 to resistor {} (ID: 3) terminal2".format(
            self.target_resistor_id
        ),
        "look",
        "answer {} ohm".format(self.target_resistance),
    ]

    return demo_actions
```

Code Sample 10: volume

```python
def get_demo_actions(self):
    demo_actions = [
        "take {}".format(self.useful_info[1].name),
        "measure the length of the {} with the {}".format(
            self.useful_info[0].name, self.useful_info[1].name
        ),
        "measure the width of the {} with the {}".format(
            self.useful_info[0].name, self.useful_info[1].name
        ),
        "measure the height of the {} with the {}".format(
            self.useful_info[0].name, self.useful_info[1].name
        ),
        "answer {} cubic cm".format(self.target_box_volume),
    ]

    return demo_actions
```

Code Sample 11: sunburn

```python
def get_demo_actions(self):
```

```python
        return [
            "use sunscreen (ID: 4)",
            "move to beach (ID: 3)",
            "look",
            "take ball (ID: 8)",
            "move to house (ID: 2)",
            "put ball (ID: 8) in box (ID: 5)",
        ]
```

Code Sample 12: space-walk

```python
    def get_demo_actions(self):
        return [
            "put on space suit (ID: 7)",
            "open inner door (ID: 5)",
            "move to airlock (ID: 3)",
            "look",
            "close inner door (ID: 5)",
            "open outer door (ID: 6)",
            "move to outer space (ID: 4)",
        ]
```

Code Sample 13: sweep-floor

```python
    def get_demo_actions(self):
        sweep_actions = []
        for garbage in self.useful_info:
            sweep_actions.append(
                "sweep {} to dustpan (ID: 3) with broom (ID: 2)".format(garbage.name)
            )

        demo_actions = (
            ["take broom (ID: 2)", "take dustpan (ID: 3)"]
            + sweep_actions
            + [
                "open garbage can (ID: 4)",
                "dump dustpan (ID: 3) to garbage can (ID: 4)",
            ]
        )

        return demo_actions
```

Code Sample 14: volume-container

```python
    def get_demo_actions(self):
        demo_actions = [
            "take {}".format(self.useful_info[0].name),
            "put {} in sink (ID: 2)".format(self.useful_info[0].name),
            "turn on sink (ID: 2)",
            "turn off sink (ID: 2)",
            "take {}".format(self.useful_info[0].name),
            "pour water in {} into {}".format(
                self.useful_info[0].name, self.useful_info[1].name
            ),
            "look",
            "answer {} mL".format(self.target_water_container_volume),
        ]

        return demo_actions
```

Code Sample 15: bath-tub-water-temperature

```python
    def get_demo_actions(self):

        water_temp = self.useful_info[0]
```

```
        cooling = [
            "turn on cold tap (ID: 5)",
            "turn off cold tap (ID: 5)",
            "use thermometer (ID: 6) on water (ID: 3)",
        ]
        hotting = [
            "turn on hot tap (ID: 4)",
            "turn off hot tap (ID: 4)",
            "use thermometer (ID: 6) on water (ID: 3)",
        ]

        if water_temp > 40:
            cooling_times = (water_temp - 35) // 5
            water_actions = cooling * cooling_times

        elif water_temp < 35:
            hotting_times = (40 - water_temp) // 5
            water_actions = hotting * hotting_times
        else:
            water_actions = []

        demo_actions = (
            [
                "take thermometer (ID: 6)",
                "use thermometer (ID: 6) on water (ID: 3)",
            ]
            + water_actions
            + ["bath"]
        )

        return demo_actions
```

Code Sample 16: conductivity

```
    def get_demo_actions(self):
        demo_actions = [
            "connect light bulb (ID: 2) terminal1 to red wire (ID: 3) terminal1",
            "connect red wire (ID: 3) terminal2 to battery (ID: 6) anode",
            "connect battery (ID: 6) cathode to black wire (ID: 4) terminal1",
            "connect black wire (ID: 4) terminal2 to fork (ID: 7) terminal1",
            "connect fork (ID: 7) terminal2 to blue wire (ID: 5) terminal1",
            "connect blue wire (ID: 5) terminal2 to light bulb (ID: 2) terminal2",
            "look",
            "take fork (ID: 7)",
        ]
        if self.useful_info[0]:
            return demo_actions + ["put fork (ID: 7) in red box (ID: 8)"]

        else:
            return demo_actions + ["put fork (ID: 7) in black box (ID: 9)"]
```

Code Sample 17: make-campfire

```
    def get_demo_actions(self):
        return [
            "take axe (ID: 4)",
            "use axe (ID: 4) on tree (ID: 5)",
            "look",
            "use axe (ID: 4) on chopped down tree (ID: 5)",
            "look",
            "take firewood (ID: 5)",
            "put firewood (ID: 5) in fire pit (ID: 2)",
            "take match (ID: 3)",
            "use match (ID: 3) on firewood (ID: 5)",
        ]
```

Code Sample 18: refrigerate-food

```python
def get_demo_actions(self):
    take_objects = []

    put_objects = []
    for food in self.useful_info:
        take_objects.append("take {}".format(food.name))
        put_objects.append("put {} in fridge (ID: 2)".format(food.name))

    demo_actions = (
        take_objects
        + ["open fridge (ID: 2)"]
        + put_objects
        + [
            "close fridge (ID: 2)",
            "look",
            "look",
            "look",
        ]
    )

    return demo_actions
```

Code Sample 19: volume-stone

```python
def get_demo_actions(self):
    demo_actions = [
        "take measuring cup (ID: 4)",
        "put measuring cup (ID: 4) in sink (ID: 2)",
        "turn on sink (ID: 2)",
        "turn off sink (ID: 2)",
        "take measuring cup (ID: 4)",
        "examine measuring cup (ID: 4)",
        "take stone (ID: 3)",
        "put stone (ID: 3) in measuring cup (ID: 4)",
        "examine measuring cup (ID: 4)",
        "answer {}".format(self.answer_volume),
    ]

    return demo_actions
```

Code Sample 20: bird-life-cycle

```python
def get_demo_actions(self):
    return [
        "sit on egg",
        "sit on egg",
        "sit on egg",
        "sit on egg",
        "sit on egg",
        "feed young bird",
        "feed young bird",
        "feed young bird",
        "feed young bird",
        "feed young bird",
    ]
```

Code Sample 21: balance-scale-weigh

```python
def get_demo_actions(self):
    weight_list = [1, 1, 2, 5, 10]
    weight_list_index = [False] * 5

    def _find_combination():
        remaining = self.cube_weight
```

```
        # using 10
        if remaining >= 10:
            weight_list_index[-1] = True
            remaining -= 10

        # using 5
        if remaining >= 5:
            weight_list_index[-2] = True
            remaining -= 5

        # using 2
        if remaining >= 2:
            weight_list_index[-3] = True
            remaining -= 2

        if remaining > 0:
            if remaining == 1:
                weight_list_index[0] = True
            if remaining == 2:
                weight_list_index[0] = True
                weight_list_index[1] = True

    _find_combination()

    weight_actions = []
    for idx, weight_list_idx in enumerate(weight_list_index):
        if weight_list_idx:
            weight_actions += [
                "take {}".format(self.useful_info[idx].name),
                "put {} in right side of the balance scale (ID: 4)".format(
                    self.useful_info[idx].name
                ),
                "look",
            ]

    demo_actions = (
        [
            "take cube (ID: 10)",
            "put cube (ID: 10) in left side of the balance scale (ID: 3)",
        ]
        + weight_actions
        + ["answer {}g".format(self.cube_weight)]
    )

    return demo_actions
```

Code Sample 22: metal-detector

```
    def get_demo_actions(self):
        agent_init_position = self.useful_info[0]
        targe_position = self.useful_info[1]

        direction = (
            targe_position[0] - agent_init_position[0],
            targe_position[1] - agent_init_position[1],
        )
        h_dir_list = (
            ["south"] * direction[0]
            if direction[0] > 0
            else ["north"] * (-direction[0])
        )
        v_dir_list = (
            ["east"] * direction[1] if direction[1] > 0 else ["west"] * (-direction[1])
        )

        dir_list = h_dir_list + v_dir_list
        self.random.shuffle(dir_list)
```

```
        detect_actions = [
            "detect with metal detector (ID: 15)",
        ]
        for dir_step in dir_list:
            detect_actions.append("move {}".format(dir_step))
            detect_actions.append("detect with metal detector (ID: 15)")

        demo_actions = (
            ["take metal detector (ID: 15)", "take shovel (ID: 16)"]
            + detect_actions
            + ["look", "dig with shovel (ID: 16)", "look", "take metal case (ID: 11)"]
        )

        return demo_actions
```

Code Sample 23: cooking

```
    def get_demo_actions(self):
        cooking_actions = [
        ]

        for cooking_item in self.receipt:
            operations = self.receipt[cooking_item]
            cooking_actions += ["take {}".format(cooking_item.name)]
            for operation in operations:
                if operation in ["slice", "dice", "chop"]:
                    cooking_actions += [
                        "{} {} with {}".format(
                            operation, cooking_item.name, self.useful_info["knife"].name
                        )
                    ]
                if operation in ["fry"]:
                    cooking_actions += [
                        "cook {} in {}".format(
                            cooking_item.name, self.useful_info["stove"].name
                        )
                    ]
                if operation in ["roast"]:
                    cooking_actions += [
                        "cook {} in {}".format(
                            cooking_item.name, self.useful_info["oven"].name
                        )
                    ]

        demo_actions = (
            [
                "take {}".format(self.useful_info["cook_book"].name),
                "read {}".format(self.useful_info["cook_book"].name),
                "take {}".format(self.useful_info["knife"].name),
            ]
            + cooking_actions
            + [
                "prepare meal",
            ]
        )

        return demo_actions
```

Code Sample 24: make-ice-cubes

```
    def get_demo_actions(self):
        return [
            "open freezer (ID: 2)",
            "examine freezer (ID: 2)",
            "take ice cube tray (ID: 3)",
            "put ice cube tray (ID: 3) in sink (ID: 4)",
            "turn on sink (ID: 4)",
```

```
        "turn off sink (ID: 4)",
        "take ice cube tray (ID: 3)",
        "put ice cube tray (ID: 3) in freezer (ID: 2)",
        "close freezer (ID: 2)",
        "look",
        "look",
        "look",
    ]
```

Code Sample 25: balance-scale-heaviest

```python
def get_demo_actions(self):

    demo_actions = [
        "take {}".format(self.useful_info[0][0].name),
        "put {} in left side of the balance scale (ID: 3)".format(
            self.useful_info[0][0].name
        ),
        "take {}".format(self.useful_info[1][0].name),
        "put {} in right side of the balance scale (ID: 4)".format(
            self.useful_info[1][0].name
        ),
        "look",
    ]

    if len(self.useful_info) == 2:
        if self.useful_info[0][1] > self.useful_info[1][1]:
            # left is heavier

            demo_actions.append("take {}".format(self.useful_info[0][0].name))
            demo_actions.append(
                "put {} in {}".format(
                    self.useful_info[0][0].name, self.answer_box.name
                )
            )
            return demo_actions
        elif self.useful_info[0][1] < self.useful_info[1][1]:
            # right is heavier
            demo_actions.append("take {}".format(self.useful_info[1][0].name))
            demo_actions.append(
                "put {} in {}".format(
                    self.useful_info[1][0].name, self.answer_box.name
                )
            )
            return demo_actions
        else:
            demo_actions.append("take {}".format(self.useful_info[0][0].name))
            demo_actions.append(
                "put {} in {}".format(
                    self.useful_info[0][0].name, self.answer_box.name
                )
            )

            demo_actions.append("take {}".format(self.useful_info[1][0].name))
            demo_actions.append(
                "put {} in {}".format(
                    self.useful_info[1][0].name, self.answer_box.name
                )
            )
            return demo_actions

    on_scale = [0, 1]
    for i in range(2, len(self.useful_info)):
        if self.useful_info[on_scale[0]][1] > self.useful_info[on_scale[1]][1]:
            demo_actions += [
                "take {}".format(self.useful_info[on_scale[1]][0].name),
                "take {}".format(self.useful_info[i][0].name),
                "put {} in right side of the balance scale (ID: 4)".format(
```

```
                        self.useful_info[i][0].name
                    ),
                    "look",
                ]
                on_scale[1] = i
            else:
                demo_actions += [
                    "take {}".format(self.useful_info[on_scale[0]][0].name),
                    "take {}".format(self.useful_info[i][0].name),
                    "put {} in left side of the balance scale (ID: 3)".format(
                        self.useful_info[i][0].name
                    ),
                    "look",
                ]
                on_scale[0] = i
        max_weight = 0
        if self.useful_info[on_scale[0]][1] > self.useful_info[on_scale[1]][1]:
            demo_actions.append("take {}".format(self.useful_info[on_scale[0]][0].name))
            demo_actions.append(
                "put {} in {}".format(
                    self.useful_info[on_scale[0]][0].name, self.answer_box.name
                )
            )
            max_weight = self.useful_info[on_scale[0]][1]
        else:
            demo_actions.append("take {}".format(self.useful_info[on_scale[1]][0].name))
            demo_actions.append(
                "put {} in {}".format(
                    self.useful_info[on_scale[1]][0].name, self.answer_box.name
                )
            )
            max_weight = self.useful_info[on_scale[1]][1]

        for cube, cube_mass in self.useful_info:
            if cube_mass == max_weight:
                demo_actions += [
                    "take {}".format(cube.name),
                    "put {} in {}".format(cube.name, self.answer_box.name),
                ]

        return demo_actions
```

Code Sample 26: take-photo

```
    def get_demo_actions(self):
        speed = self.camera.properties["current_shutter_speed"]
        iso = self.camera.properties["current_iso"]
        aperture = self.camera.properties["current_aperture"]

        target_aperture = self.useful_info[0]
        target_speed = self.useful_info[1]
        target_iso = self.useful_info[2]

        adjust_actions = [
            "focus {}".format(self.target_food.name),
        ]
        if target_aperture > aperture:
            adjust_actions += ["rotate aperture clockwise"] * (
                target_aperture - aperture
            )
        elif target_aperture < aperture:
            adjust_actions += ["rotate aperture anticlockwise"] * (
                -target_aperture + aperture
            )
        else:
            pass
        if target_speed > speed:
            adjust_actions += ["rotate shutter speed clockwise"] * (
```

```
                target_speed - speed
            )
        elif target_speed < speed:
            adjust_actions += ["rotate shutter speed anticlockwise"] * (
                -target_speed + speed
            )
        else:
            pass
        if target_iso > iso:
            adjust_actions += ["rotate iso clockwise"] * (target_iso - iso)
        elif target_iso < iso:
            adjust_actions += ["rotate iso anticlockwise"] * (-target_iso + iso)
        else:
            pass

        demo_actions = ["take camera (ID: 2)"] + adjust_actions + ["press shutter"]

        return demo_actions
```

Code Sample 27: plant-tree

```
    def get_demo_actions(self):
        return [
            "take shovel (ID: 2)",
            "dig with shovel (ID: 2)",
            "take tree (ID: 8)",
            "look",
            "put tree (ID: 8) in hole (ID: 9)",
            "inventory",
            "put soil (ID: 10) in hole (ID: 9)",
            "take jug (ID: 7)",
            "put jug (ID: 7) in sink (ID: 5)",
            "turn on sink (ID: 5)",
            "turn off sink (ID: 5)",
            "take jug (ID: 7)",
            "pour water in jug (ID: 7) into soil (ID: 10)",
        ]
```

Code Sample 28: boil-water

```
    def get_demo_actions(self):
        return [
            "take pot (ID: 4)",
            "put pot (ID: 4) in sink (ID: 3)",
            "examine sink (ID: 3)",
            "turn on sink (ID: 3)",
            "examine sink (ID: 3)",
            "turn off sink (ID: 3)",
            "take pot (ID: 4)",
            "look",
            "put pot (ID: 4) on stove (ID: 2)",
            "examine stove (ID: 2)",
            "turn on stove (ID: 2)",
            "examine stove (ID: 2)",
            "examine stove (ID: 2)",
            "examine stove (ID: 2)",
        ]
```

Code Sample 29: forge-key

```
    def get_demo_actions(self):
        demo_actions = [
            "take copper ingot (ID: 4)",
            "put copper ingot (ID: 4) in foundry (ID: 3)",
            "turn on foundry (ID: 3)",
            "look",
            "look",
```

```
            "look",
            "look",
            "look",
            "look",
            "pour copper (liquid) (ID: 4) into key mold (ID: 6)",
            "look",
            "look",
            "take copper key (ID: 4)",
            "open door (ID: 5) with copper key (ID: 4)",
        ]

        return demo_actions
```

Code Sample 30: inclined-plane

```
    def get_demo_actions(self):
        look_table = {
            0.5: ["look"] * 5,
            1: ["look"] * 5,
            1.5: ["look"] * 5,
            2: ["look"] * 5,
        }

        a1 = self.useful_info[0][0]
        a2 = self.useful_info[0][1]
        a1_look = look_table[a1]
        a2_look = look_table[a2]

        demo_actions = (
            [
                "take stopwatch (ID: 5)",
                "take block (ID: 4)",
                "put block (ID: 4) on inclined plane 1 (ID: 2)",
                "activate stopwatch (ID: 5)",
            ]
            + a1_look
            + [
                "deactivate stopwatch (ID: 5)",
                "examine stopwatch (ID: 5)",
                "reset stopwatch (ID: 5)",
                "take block (ID: 4)",
                "put block (ID: 4) on inclined plane 2 (ID: 3)",
                "activate stopwatch (ID: 5)",
            ]
            + a2_look
            + [
                "deactivate stopwatch (ID: 5)",
                "examine stopwatch (ID: 5)",
            ]
        )
        if a1 > a2:
            return demo_actions + ["focus on inclined plane 2 (ID: 3)"]
        else:
            return demo_actions + ["focus on inclined plane 1 (ID: 2)"]
```

Code Sample 31: wash-clothes

```
    def get_demo_actions(self):
        washing, drying, busketing = [], [], []

        for cloth in self.dirty_clothes:
            washing += [
                "take {}".format(cloth.name),
                "put {} in washing machine (ID: 2)".format(cloth.name),
            ]

            drying += [
```

```
                "take {}".format(cloth.name),
                "put {} in dryer (ID: 3)".format(cloth.name),
            ]

            busketing += [
                "take {}".format(cloth.name),
                "put {} in basket (ID: 12)".format(cloth.name),
            ]

        for cloth in self.clean_clothes:
            busketing += [
                "take {}".format(cloth.name),
                "put {} in basket (ID: 12)".format(cloth.name),
            ]

        demo_actions = (
            [
                "open washing machine (ID: 2)",
            ]
            + washing
            + [
                "use bottle of detergent (ID: 4) on washing machine (ID: 2)",
                "close washing machine (ID: 2)",
                "turn on washing machine (ID: 2)",
                "wait",
                "look",
                "look",
                "open washing machine (ID: 2)",
                "open dryer (ID: 3)",
            ]
            + drying
            + [
                "close dryer (ID: 3)",
                "turn on dryer (ID: 3)",
                "wait",
                "look",
                "look",
                "open dryer (ID: 3)",
                "take skirt (ID: 5)",
            ]
            + busketing
        )

        return demo_actions
```

## B.3 Analysis of Demo Actions

Figure 8 displays the numbers of steps of the generated rule-based policies for environments to complete the tasks. We note that the number of steps may vary due to the randomness in the environments. For example, in mix paint, if the target color is black, 3 steps are needed, and other colors may only require 2 steps.

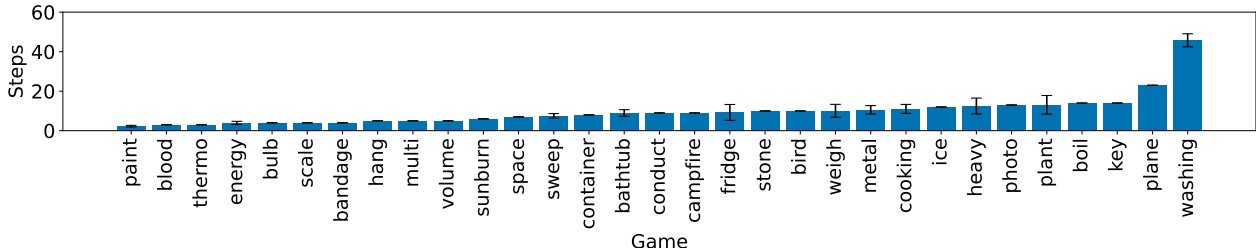

Figure 8: Steps to complete the tasks

# C  Prompts for World Model

## C.1  Prompt for Next State and Reward/Terminal Predictions

Code Sample 32: Code for Prompts of Generating Potential Actions.

```python
prompt = (
        "You are a simulator of a text game. Read the task description of a text game. "
        "Given the current game state in JSON, "
        "you need to decide the new game state after taking an action including the game
    score.\n"
    )
    prompt += (
        "Your response should be in the JSON format. "
        "It should have three keys: 'modified', 'removed', and 'score'. "
        "The 'modified' key stores a list of all the object states that are added or
    changed after taking the action. "
        "Keep it an empty list if no object is added or modified. "
        "The 'removed' key stores a list of uuids of the objects that are removed. "
        "Keep it an empty list if no object is removed. "
        "The 'score' key stores a JSON with three keys: "
        "'score', 'gameOver', and 'gameWon'. "
        "'score' stores the current score, "
        "'gameOver' stores a bool value on whether the game is over, "
        "and 'gameWon' stores a bool value on whether the game is won. \n"
    )

    last_action = "" if len(self.last_actions) == 0 else self.last_actions[-1]
    max_UUID = importlib.import_module(self.game_name).UUID
    if current_state is None:
        current_state = get_state(self.game, last_action, max_UUID, self.game_name)
        current_state_for_prompt = make_game_state(current_state)
        max_uuid = current_state["max_UUID"]
    else:
        # print("use the predicted state")

        current_state_for_prompt = current_state
        max_uuid = len(current_state["game_state"])

        # start adding examples
    example_prompt = self.build_examples()
    prompt += example_prompt
    # end of adding examples
    # Task
    prompt += "Here is the game that you need to simulate:\n"
    prompt += "Task Description:\n"
    prompt += f"{self.task_desc}\n"

    # load rules
    obj_desc = preprocess_obj_desc(self.obj_rules[self.game_name])
    action_desc = self.action_rules[self.game_name]
    score_desc = self.score_rules[self.game_name]

    prompt += "Here are the descriptions of all game objects properties:\n"
    prompt += obj_desc.strip()
    prompt += "\n"
    prompt += "Here are the descriptions of all game actions:\n"
    prompt += action_desc.strip()
    prompt += "\n"
    prompt += "Here is a description of the game score function:\n"
    prompt += score_desc.strip()
    prompt += "\n"

    # data_state, data_UUID_base, data_action = None, None, None
    prompt += "Here is the game state:\n"
    prompt += f"{current_state_for_prompt}\n"
    prompt += "\n"
```

```
        prompt += f"The current game UUID base is {max_uuid}\n"
        prompt += f"The action to take is:\n{action}\n"
```

Code Sample 33: The Predicted Next State and Reward/Terminal in lit-lightbulb (bulb) environment.

```
{'game_state': [{'name': 'room (ID: 1)', 'uuid': 1, 'type': 'World', 'properties': {'
    isContainer': True, 'isMoveable': True, 'isOpenable': False, 'isOpen': True, '
    containerPrefix': 'in'}, 'contains': ['agent (ID: 0)', 'light bulb (ID: 2)', 'red wire (
    ID: 3)', 'black wire (ID: 4)', 'blue wire (ID: 5)', 'battery (ID: 6)']}, {'name': 'agent
     (ID: 0)', 'uuid': 0, 'type': 'Agent', 'properties': {'isContainer': True, 'isMoveable':
     True, 'isOpenable': False, 'isOpen': True, 'containerPrefix': 'in'}, 'contains': []}, {
    'name': 'light bulb (ID: 2)', 'uuid': 2, 'type': 'LightBulb', 'properties': {'
    isContainer': False, 'isMoveable': True, 'is_electrical_object': True, 'conductive':
    True, 'connects': {'terminal1': [3, 'terminal1'], 'terminal2': [None, None]}, 'on':
    False}, 'contains': []}, {'name': 'red wire (ID: 3)', 'uuid': 3, 'type': 'Wire', '
    properties': {'isContainer': False, 'isMoveable': True, 'is_electrical_object': True, '
    conductive': True, 'connects': {'terminal1': [2, 'terminal1'], 'terminal2': [None, None
    ]}, 'is_wire': True}, 'contains': []}, {'name': 'black wire (ID: 4)', 'uuid': 4, 'type':
     'Wire', 'properties': {'isContainer': False, 'isMoveable': True, 'is_electrical_object'
    : True, 'conductive': True, 'connects': {'terminal1': (None, None), 'terminal2': (None,
    None)}, 'is_wire': True}, 'contains': []}, {'name': 'blue wire (ID: 5)', 'uuid': 5, '
    type': 'Wire', 'properties': {'isContainer': False, 'isMoveable': True, '
    is_electrical_object': True, 'conductive': True, 'connects': {'terminal1': (None, None),
     'terminal2': (None, None)}, 'is_wire': True}, 'contains': []}, {'name': 'battery (ID:
    6)', 'uuid': 6, 'type': 'Battery', 'properties': {'isContainer': False, 'isMoveable':
    True, 'is_electrical_object': True, 'conductive': True, 'connects': {'cathode': (None,
    None), 'anode': (None, None)}}, 'contains': []}]}, {'score': 0, 'gameOver': False, '
    gameWon': False}
```

## C.2  Prompts of Generating Potential Actions.

Code Sample 34: Code for Prompts of Generating Potential Actions.

```
        prompt = (
            "You are a simulator of a text game. "
            "Read the task description and the descriptions of all game actions of a text
    game. "
            "Given the current game state in JSON, and the previous actions that lead to the
     current game state, "
            "you need to decide the most {} actions "
            "that can help to complete the task step by step at the current state.\n".format
    (
                k
            )
        )
        prompt += (
            "Each of your action should in one phrase with one verb and the objects it
    operates on. "
            "Examples of actions includes:\n"
            "move south" + ",\n"
            "detect with metal detector (ID: 15)" + ",\n"
            "dig with shovel (ID: 16)" + ",\n"
            "open freezer (ID: 2)" + ",\n"
            "put ice cube tray (ID: 3) in sink (ID: 4)" + ",\n"
            "dice patato (ID: 2) with knife (ID: 8)" + ",\n"
            "give Type O negative blood (ID: 3) to patient (ID: 2)" + ",\n"
            "read cook book (ID: 7)" + ".\n"
        )

        prompt += (
            "Your response should be in the JSON format. "
            "It should have one key: 'avail_actions', which includes the list of the
    recommended actions. \n"
        )

        last_action = "" if len(self.last_actions) == 0 else self.last_actions[-1]
```

```
        max_UUID = importlib.import_module(self.game_name).UUID
        if current_state is None:
            current_state = get_state(self.game, last_action, max_UUID, self.game_name)
            current_state_for_prompt = make_game_state(current_state)
            max_uuid = current_state["max_UUID"]
        else:
            # print("use the predicted state")

            current_state_for_prompt = current_state
            max_uuid = len(current_state["game_state"])

            # start adding examples
        # example_prompt = self.build_examples()
        # prompt += example_prompt
        # end of adding examples
        # Task
        prompt += "Here is the game that you need to simulate:\n"
        prompt += "Task Description:\n"
        prompt += f"{self.task_desc}\n"

        # load rules
        obj_desc = preprocess_obj_desc(self.obj_rules[self.game_name])
        action_desc = self.action_rules[self.game_name]
        score_desc = self.score_rules[self.game_name]

        prompt += "Here are the descriptions of all game objects properties:\n"
        prompt += obj_desc.strip()
        prompt += "\n"
        prompt += "Here are the descriptions of all game actions:\n"
        prompt += action_desc.strip()
        prompt += "\n"
        prompt += "Here is a description of the game score function:\n"
        prompt += score_desc.strip()
        prompt += "\n"

        # data_state, data_UUID_base, data_action = None, None, None
        prompt += "Here is the game state:\n"
        prompt += f"{current_state_for_prompt}\n"
        prompt += "\n"

        prompt += f"The current game UUID base is {max_uuid}\n"

        if len(self.last_actions) == 0:
            prompt += "There is no previous actions."
        else:
            prompt += "The previous actions {}:\n".format(
                "is" if len(self.last_actions) == 1 else "are"
            )
            for action in self.last_actions:
                prompt += action + "\n"
```

Code Sample 35: The Generated Actions of make-ice-cubes (ice) environment.

```
{'avail_actions': ['open freezer (ID: 2)', 'take ice cube tray (ID: 3) from freezer (ID: 2)'
    , 'put ice cube tray (ID: 3) in sink (ID: 4)', 'turn on sink (ID: 4)', 'take ice cube
    tray (ID: 3) from sink (ID: 4)', 'put ice cube tray (ID: 3) in freezer (ID: 2)', 'close
    freezer (ID: 2)', 'wait for ice to form', 'open freezer (ID: 2)', 'check ice cube tray (
    ID: 3) for ice']}
```

# D   Accuracy of Policy Verification

We provide the accuracy of the policy verification regarding the three criteria, i.e., score, gameWon and gameOver. We note that the performance on gameWon and gameOver predictions are far better than the prediction of score.

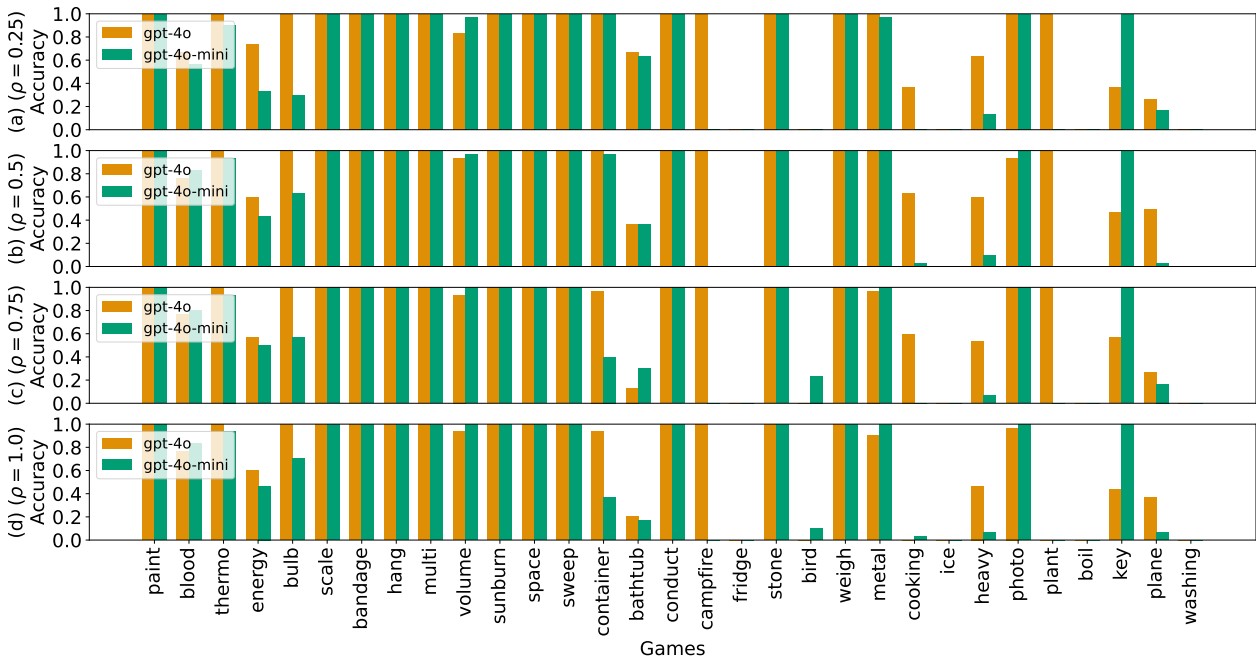

Figure 9: The accuracy of the world model to verify the correct policies on score

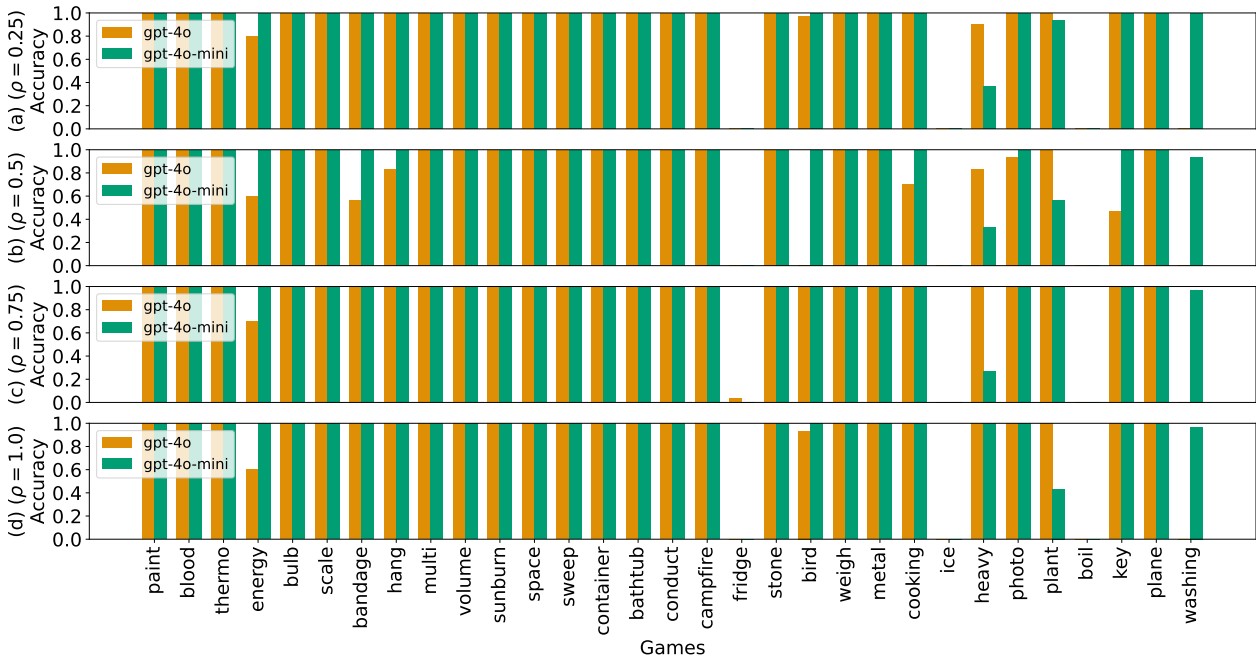

Figure 10: The accuracy of the world model to verify the correct policies on gameOver

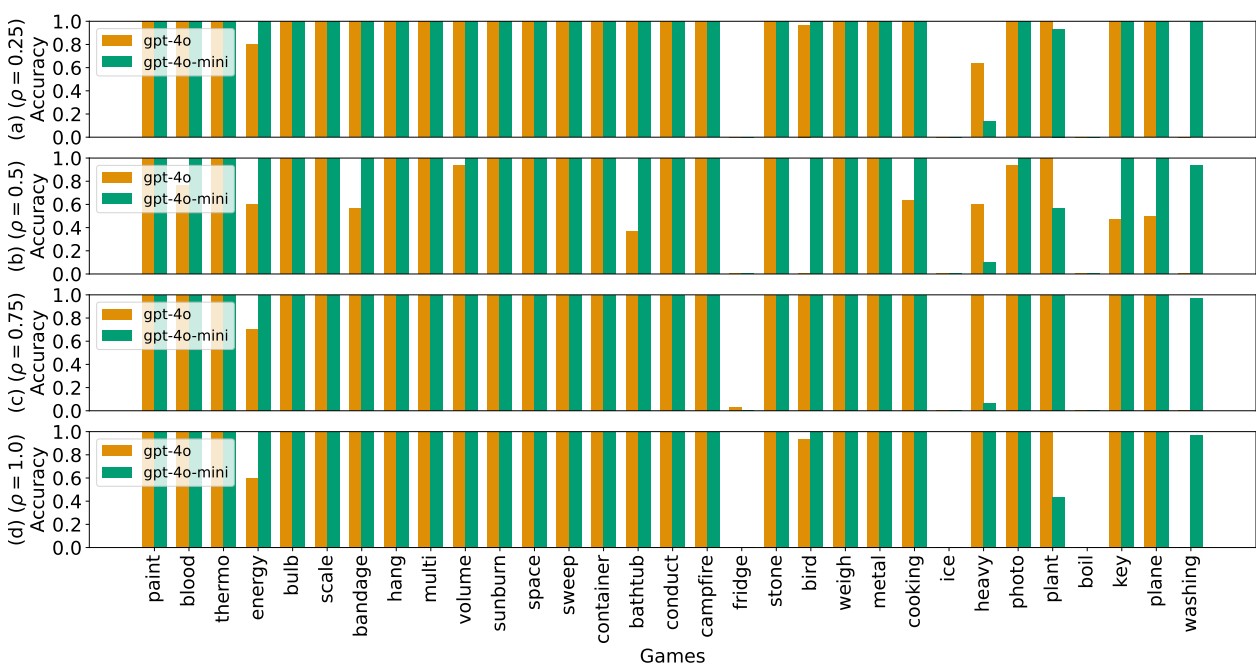

Figure 11: The accuracy of the world model to verify the correct policies on gameWon

# E    Step Accuracy of Action Proposal

We also provide the accuracy of each steps for the action proposal tasks. We observe for most of the task, there is some key steps that the world model has low accuracy for the action proposal, which brings difficulties for the world model to complete the tasks.

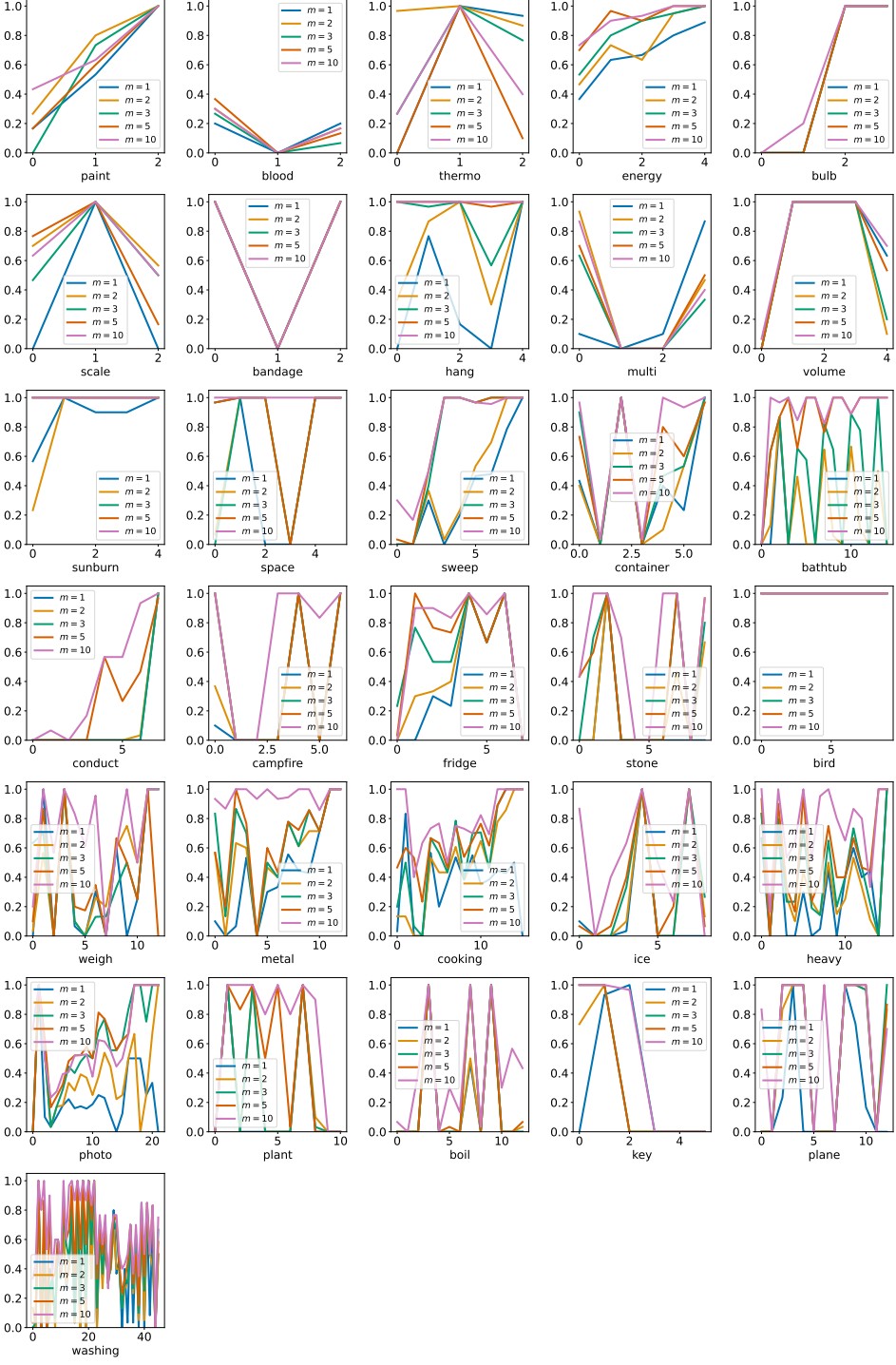

Figure 12: Step correctness of the action proposal of GPT-4o-mini

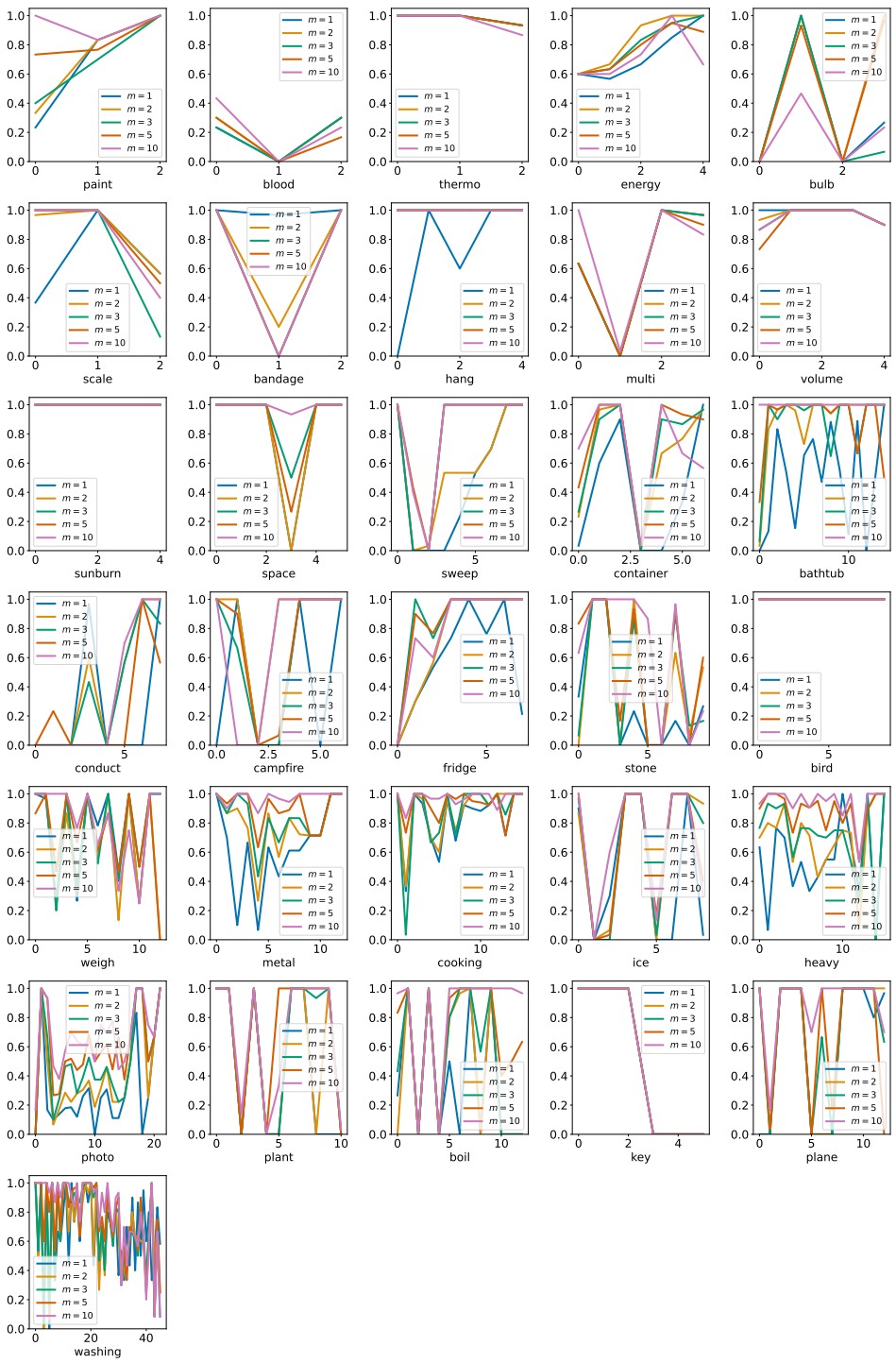

Figure 13: Step correctness of the action proposal of GPT-4o

