# OpenReview forum: "Evaluating World Models with LLM for Decision Making"
_TMLR — Rejected by TMLR_

### Review · Reviewer_EE4M · 2024-12-10

**Summary Of Contributions:**

This work proposes to evaluate world models based on 3 specific tasks:
- Policy verification: the model predicts the next state and reward given the current state and action taken from the golden policy.
- Action proposal: the model predicts the next possible action(s) based on the current state.
- Policy planning: the model predicts both the next action (top 1) and the resulting next state and reward.

This paper evaluates GPT-4o and GPT-4o-mini  on the above three tasks using 31 text world environments generated from the ByteSized32 benchmark.
The main observations are:
- GPT-4o significantly outperforms GPT-4o-mini
- The performance is lower for long-horizon tasks
- Making the model predict the sequence of next actions and states is unstable.

**Audience:**

Yes

**Broader Impact Concerns:**

N.A.

**Claims And Evidence:**

Yes

**Requested Changes:**

A major rewrite of the paper for a better understanding. There is not a lot of claims in the paper so I think the claims are matched with evidence but the narrative is not clear.

A more detailed analysis on the type of success and failures for each model to share valuable findings. Are there common failures that arise in particular tasks / models? What can be done to try to solve them? What are the models good at? In the Action prediction task, what is the proportion of times you need to use the closest match embedding? Another solution would be to list all possible candidate actions in the model prompt.

**Strengths And Weaknesses:**

**Strengths**

The paper correctly identifies two tasks that world models can perform: next-state prediction and next-action prediction. It also explores running one task after another to make the model predict entire trajectories.
Experiments show the performance of GPT4o and GPT4o-mini as world models on a set of 31 text world environments.

**Weaknesses**

**W1** The contribution of this work is not clear. The three tasks defined (next state prediction, next action prediction and their combination - named 'policy verification', 'action proposal', and 'policy planning’ respectively in the paper) are standard practices in many previous works [[1](https://arxiv.org/abs/1803.10122),[2](https://arxiv.org/abs/2402.15391),[3](https://arxiv.org/abs/2010.02903),[4](https://arxiv.org/abs/2302.05507)].

**W2** The experimental observations do not bring substantial findings.
- GPT4o is expected to perform lower than GPT4 (https://openai.com/index/gpt-4o-mini-advancing-cost-efficient-intelligence/)
- Long-horizon tasks are often more challenging than short-horizon ones simply due to the fact that there are more opportunities for the LLM to make a mistake.
- Letting a model simulate both an agent acting in an environment **and** the environment response is challenging [[4](https://arxiv.org/abs/2302.05507)]. Doing this over multiple time steps will be unstable due to error propagation.

**W3** Eventually, the writing is very weak and hard to understand. The paper contains many typos.

Sentences that fit on 5 lines should be rephrased into smaller sentences for easier parsing. Ex:
> _First, the objective of decision making is finding the policy to complete the task, where only a small portion of the world will be visited during finding and executing the policy, given the fact that AlphaZero can find the super-human policy (Silver et al., 2018) by only exploring a small proportion (less than 1%) of the state space, therefore, the evaluation of the world models on the predictions of the transitions that relevant to the desired policy is more important than the transitions far from the policy._

Figure 1 is not clear. In particular, it is not clear why “_The prediction of B in Case 2 leads to the wrong action_”.

The paper contains contradicting statements:

Section 1:
> Several works (Xie et al., 2024; Janner et al., 2021) incorporate the action selection into consideration.

Section 4:
> The action proposal is a novel task for world model, based on Observation 2, which is not considered in previous work.

---

[1] World Models; David Ha, Jürgen Schmidhuber; https://arxiv.org/abs/1803.10122

[2] Genie: Generative Interactive Environments; Bruce et. al; https://arxiv.org/abs/2402.15391

[3] Keep CALM and Explore: Language Models for Action Generation in Text-based Games; Yao et al.; https://arxiv.org/abs/2010.02903

[4] Language Decision Transformers with Exponential Tilt for Interactive Text Environments; Gontier et. al; https://arxiv.org/abs/2302.05507

---

> ### Author Response · Authors · 2025-01-11
> **Response to EE4M**
>
> Thanks for your helpful comments. Below are our responses.
>
> 1.	About the contribution and the objective of this paper. We respectively do not agree that the policy verification, action proposal and policy planning is a common practice in the current world model literature, especially for the world models with LLMs. We argue that the world model is usually evaluated as i) general world simulators, e.g., Genie, or ii) additional modules for the decision making where an actor is trained, e.g., MuZero, Dreamer, and the world model in (Ha et al. 2018). For the references 3 and 4 you referred, it is more about the actors selecting the actions, rather than the world models. We should distinguish the world model from the actors and evaluate the world model for decision-making independently, which is the primary objective of this work. WE add more justifications about the tasks and the objective in Appendix A.
>
> 2.	About the model. We do not use GPT-4 due to the high cost. And GPT-4o and GPT-4o-mini are two most-widely used models with advanced capabilities.
>
> 3.	About Figure 1. We should pick the action with the higher value. The prediction of the A’s value in Case 1 is less accurate than Case 2, but it leads to the correct action. Therefore, more accurate predictions do not necessarily lead to better decisions.
>
> 4.	About the findings. We rewrite the findings in the paper and provide a deeper analysis in the paper. Specifically, we observe that more steps to complete the tasks do not necessarily be more difficult to solve, which differs from the traditional RL methods where exploration is needed.
>
> 5.	About the action proposal. Given the recommended actions generated by the LLMs, we need to retrieve the most relevant actions to execute with the text embedding models. For the suggestion of adding all actions into the prompts, there are more than 500 available actions in most of the timesteps, sometimes more than 1500 available actions, which may be too long to add all available into the prompt. On the other hand, for the tasks such as web browsing, there are infinite number of actions, and we cannot enumerate all the actions. Therefore, we add the action rules into the prompts and ask the LLMs to generate the actions.
>
> 6.	For the failures, as we discussed in the paper, we find that the LLMs can complete the daily tasks, which usually requires the common knowledge, while the LLMs usually fails in the scientific tasks, which indicates that the LLMs may lack the scientific knowledge. The methods to fix these issues include in-context learning, retrieval-augmented generation and fine-tuning, and we will include these methods into the evaluation in the future work.
>
> 7.	We have revised the paper to avoid the contradictions. Specifically, Trajectory Transformer (Janner et al., 2021) trains the transformer model to generate both the next-state prediction and the action, which is not related to LLMs. (Xie et al., 2024) only use the LLMs to valid the actions and does consider the action proposal.

---

> ### Comment · Reviewer_EE4M · 2025-01-16
> **response to authors**
>
> Thank you for your response. I have some followup questions.
>
> 1. Can you explain precisely what is the difference between "_evaluated as i) general world simulators,_" and your "_policy planning_" task. My understanding is that in both cases the model estimates the following distribution $P(s_{t+1} | a_t, s_t) * P(a_t | s_t)$
> 2. sorry for the typo, I meant to write gpt4o and gpt4o-mini. my first point to W2 should be read as: "_GPT4o-mini is expected to perform lower than GPT4o_"
> 3. I still have trouble understanding this figure. What are "A" and "B"? are those two candidate actions at a specific timestep? I can see that the "Case2" bar is closer to the "True Value / B" bar than "Case1". But I don't see why it leads to the wrong action. What is "_the prediction of the A’s value in Case 1_"? I only see the prediction of B? Same for Case2.
> 4. Thanks for the insight, indeed domain knowledge plays an important role here.
> 5. thanks for clarifying. indeed 1500 candidate actions will not fit in the prompt. It is nonetheless weird that the llm sometimes generates invalid actions even if you give the action rules in the prompt. Do you know how often this happens?
> 6. thanks for clarifying.
> 7. Thanks for removing the contradiction, I updated my rating for claims supported. However, note that Trajectory Transformer (Janner et al., 2021) trains the transformer model to generate both the next-state prediction and the action, which is exactly the same as your "_policy planning_" task. The only difference is that you use a pre-trained model and a text environment while the original Trajectory Transformer uses non pre-trained models and images. LDTs with exponential tilt (Gontier et al., 2023) also generate both the next-state prediction and the action but this time in text games. All these works use the same "evaluation" method. Switching to a pretrained model or the environment does not change the mathematical property of the evaluation (see point #1).

---

> > ### Author Response · Authors · 2025-01-19
> > **Thanks for your comments**
> >
> > 1.	For the difference of general simulator and policy planning. For the general simulator, the world model need to predict every transitions of the game. Taking the Go as an example, the general simulator may need to predict the state changes for the every possible action. Therefore, the state space of Go is extremely large (10^170). However, for the decision making, only a small proportion of the state is useful, that is why AlphaGo Zero can achieve superhuman performance by exploring only a small proportion of the state (<1%). This is briefly introduced in the Introduction section of the paper. So, we argue that the world model for decision making should be evaluated in a more decision-oriented perspective, i.e., more focusing on the states relevant to the targeted policies rather than the general world simulator.
> >
> > 2.	For the two models, additional to benchmark whether GPT-4o is better than GPT-4o-mini, we would also want to test whether the GPT-4o-mini can also perform well in these decision-making tasks, given that GPT-4o-mini is also widely use due to the low cost and the acceptable performance on most daily tasks. Therefore, we choose both models. This is similar to Wang et al, 2024, where GPT-4 and GPT-3.5-turbo are the two selected models.
> >
> > 3.	About the example. This example is basically to motivate that the more accurate prediction does not lead to correct action. Given the two action A and B, we may need to predict the Q values of them and select the best one and the true values of A and B are 80 and 100, therefore we should choose B. Suppose that we can predict the A’s value accurately. Then, we have two predictions of B: 200 for Case 1 and 70 for Case 2. We have more accurate prediction in Case 2, i.e., (200-100)^2 > (70-100)^2 if measured with Euclid distance, where we will choose A, which is wrong. Therefore, better prediction does not lead to better actions.
> >
> > 4.	About the failures of LLM’s generation. The instruction following capabilities of both GPT-4o and GPT-4o-mini, where we believe that giving the rules and inferring the action to take would be a more difficult task than the common instruction following task, e.g., “write a paragraph about AI”. We primarily observe two failures: i) the LLM may predict an empty set of the states and reward/terminal, and ii) the LLM may generate wrong actions. The frequency of these failures also depends on the environments, where the "hang-painting", "space-walk", and "make-campfire" are the three environments we experiences the failures.
> >
> > 5.	Comparison with decision transformer and trajectory transformer. Thanks for point out this.
> > - i) the basic idea of the decision transformer, as well as the language decision transformer, is generating the actions conditional on the history. Though the state prediction is used to training in LDT to accelerate the training, it is not used for making decisions, where LDT only uses the model to generate the actions directly. So we do not think this is related to world model.
> > - ii) we think the trajectory transformer is more related to the world model where the state and reward prediction is used to making decisions and the planning, e.g., beam search, is used. However, TT only consider the continuous control, and is domain specific and problem specific, similar to all world model works which do not use LLMs.
> > - iii) Recently, LLMs provide a promising way to build the general world model and the world model with LLMs emerge as a novel research field. However, most of these work focus on the single-step prediction and a comprehensive evaluate is needed. We admit that this work is inspired by the insights from TT and we extends it to the world models with LLMs. Specifically, we consider the policy verification, the action proposal and the policy planning tasks, where the TT combines these tasks to generate the actions and only investigate the performance for the decision. Instead of only considering the performance of the decision makings, our three tasks provide a more bottom-up analysis of the world model for decision making.
> > - iv) To avoid any confusion, we add the comparison with DT and TT in Appendix A4, highlighted as purple.

---

> > > ### Comment · Reviewer_EE4M · 2025-01-20
> > > **Thanks for clarifications**
> > >
> > > Thank you for your reply.
> > >
> > > 1. I think I start to see the difference! Thank you for explaining this. You should maybe rephrase that part of the paper to make it more clear what the difference is. You could say something along the lines of:
> > > > General world simulators need to estimate the state transition function from **any** state $s \in S$. However in many environments, only a small portion of the state space will be visited ($S_{visit} << S$). As such we argue that LLMs as decision making world simulators should be evaluated in a more decision-oriented perspective, i.e., focusing more on the states relevant to the task at hand ($S_{visit}$).
> > >
> > > 2. of course, no problem regarding the choice of gpt4o and 4o-mini. I am just saying that finding that gpt4o is better than gpt4o-mini is not surprising.
> > >
> > > 3. Thank you for clarifying. I think I also start to understand this figure! I strongly suggest the following modifications to make it more clear for future readers:
> > > - add the estimation of action A in Case1 and Case2. Even if it is the same value as in "True Value".
> > > - rephrase the caption to something like:
> > > > In Case 1 the estimation of action B's **value** is worse than in Case 2 (compared to the True Value). However, in Case 1 the most valuable action is B, which is also the case in the True Value setting. This shows that more accurate predictions (case 2) do not always lead to correct decisions (case 1).
> > >
> > > 4. Thank you, this is exactly the type of analysis that should be further explored in the paper:
> > > > _ii) the LLM may generate wrong actions. The frequency of these failures also depends on the environments, where the "hang-painting", "space-walk", and "make-campfire" are the three environments we experiences the failures._

---

> > > > ### Author Response · Authors · 2025-01-21
> > > > **Thanks for your insightful comments**
> > > >
> > > > Dear Reviewer EE4M,
> > > >
> > > > Many thanks for your insightful comments. We have included all these valuable suggestions into the revised version of the paper.
> > > >
> > > > Best,
> > > > Authors

---

### Review · Reviewer_XpSL · 2024-12-17

**Summary Of Contributions:**

This paper proposes a suite of tasks to evaluate LLMs acting as world models for decision making tasks. Compared to prior work, the focus of the evaluation protocol is to understand how helpful these LLM-based world models can be used for obtaining a good policy, instead of simply measuring how well the predicted next states align with the ground truth. Based on this principle, they focus on 3 tasks: policy verification, action proposal, and policy planning. They conduct experiments on 31 environments, with GPT-4o and GPT-4o-mini models. They present a comprehensive evaluation varying the length of action sequence to verify, environment complexity, etc., showing that GPT-4o is generally better than GPT-4o-mini, especially when increasing the proportion of the action sequence to verify.

**Audience:**

Yes

**Broader Impact Concerns:**

No concerns.

**Claims And Evidence:**

Yes

**Requested Changes:**

1. Provide more analysis on whether the 3 tasks proposed in this work can lead to different conclusions compared to existing tasks that mostly consider next-state prediction.

2. Add more LLMs for evaluation, e.g., Gemini, Claude, and open-source LLMs.

3. Since the temperature is set to be 0, explain why is it necessary to have 30 runs in each environment.

4. Present the average and std of experimental results.

5. Explain how action planning results depend on policy verification and action proposal. See the Weaknesses section for a concrete example.

**Strengths And Weaknesses:**

Strengths:

1. The 3 tasks designed in this work are motivated well with the observations discussed in the introduction. They provide another angle to evaluate the usefulness of LLM-based world models.

2. The evaluation includes a diverse set of environments, and there is a comprehensive analysis on the performance in different settings.

3. The paper provides a very detailed description to explain the implementation details.

Weaknesses:

1. I can understand the motivation of tasks designed in this work. However, from the perspective of comparing different LLM-based world models, I cannot see whether the 3 tasks proposed in this work can lead to different conclusions compared to existing tasks that mostly consider next-state prediction. One way to improve the work is to add more LLMs for evaluation, e.g., Gemini, Claude, and open-source LLMs. Evaluating more capable LLMs might help better justify the value of new proposed tasks.

2. In Section 6, the authors mention that the temperature is set to be 0. In this case, why is it necessary to have 30 runs in each environment? Is it due to the randomness of the dynamics? Also, it will be helpful to present the average and std of experimental results.

3. Some results on the policy planning task are hard to interpret. For example, for the Stone task, both GPT-4o and GPT-4o-mini get 100% accuracy on policy verification, and GPT-4o outperforms GPT-4o-mini on action proposal. However, GPT-4o-mini outperforms GPT-4o on policy planning. Since policy planning depends on policy verification and action proposal, why GPT-4o-mini becomes better on policy planning in this case?

---

> ### Author Response · Authors · 2025-01-11
> **Response to XpSL**
>
> Thanks for your valuable comments. Below are our responses.
>
> 1.	About the proposed new tasks vs. one-step next-state prediction. We provide an illustrative example in Figure 1 to demonstrate that more accurate predictions do not necessarily lead to better decisions. Besides, most of the decision-making tasks involves multiple steps, and the one-step prediction is not suitable. Therefore, the novel evaluation tasks are needed. We hope this can help to clarify about the necessity of the newly proposed tasks. We also add more justifications in Appendix A.1.
>
> 2.	About the selection of the backbone models. For more closed-sourced models, as we need to evaluate the three novel tasks over 31 environments, and each with 30 runs, we roughly use 3000 dollars for all the experiments for GPT-4o and GPT-4o-mini. Due to the limited budget, we cannot afford to test on Claude and Gemini. For the open-sourced models, we test on some open-sourced models, e.g., Qwen 7B model, and find that current open-sourced models still cannot generate the responses with correct formats, i.e., JSON. This brings difficulties for the evaluation. Therefore, we only consider the two GPT models, given that GPT models are most widely used models with advanced instruction following capabilities. We note that this is similar to the models selected in (Wang et al., 2024). We add this justification in Appendix A.3.
>
> 3.	About the temperature and the 30 runs. We set the temperature to be 0 to reduce the randomness of the responses generated by GPT-4o and GPT-4o-mini. The 30 runs are primarily following the standards in the RL literature due to the randomness in the environments. For example, in the mix-paint tasks, the target colors may be different and the number of steps to complete the tasks is different. The error bar in Figure 4 reflects this.
>
> 4.	Error bars in Figures 5, 6, and 7. Given that the results of both the policy verification and the policy planning can only be 0 or 1, it is not necessary to plot the error bars on Figures 5 and 7. As the action proposal takes the average over the steps, the results of one run is a real value, therefore, we add the error bars on Figure 6. We add the footnotes to explain this.
>
> 5.	Results of the policy planning. We roughly observe that the better of the results on policy verification and the action proposal, the better the performance of the policy planning. However, for a specific case, e.g., stone, there are many other issues influence the real performance. For example, if the GPT-4o on average has a better performance on the action proposal, but perform very bad on some specific step, e.g., 0. This means that GPT-4o potentially never complete the task. So we want to highlight that this results are reasonable and the fully understanding of the interplay between policy verification and the action proposal requires for a specific case need further analysis and our work is just a preliminary attempt for this.
>
> 6.	We argue that analyzing the performance of the decision-making tasks is extremely complicated, our tasks is a preliminary attempt to analyze this from a bottom-up way. We believe that this paper can help people to treat the world model as the core module for the decision making and foster the understanding and development of novel and advanced world models.

---

### Review · Reviewer_xu2s · 2024-12-29

**Summary Of Contributions:**

The paper presents a benchmarking effort of LLMs as world models, specifically their ability to predict the state, reward, terminality, and/or action of an MDP in various environments.

GPT-4o and GPT-4o-mini are benchmarked on three tasks: policy verification, in which they are tested on their ability to predict state, reward, and terminality given actions in sequence; action proposal, in which they choose actions in various lengths of sequence and with various amounts of input; and policy planning, in which they do both, building on their own predictions on top of some initial subtrajectory.

They are tested on 31 environments of varied length and overlap with 4o and 4o-mini's domains.

The paper lists three key findings:
1. 4o is better than 4o-mini, particularly in terms of how many tasks are within its domain
2. Longer action sequences are harder to generate
3. Combining functionalities - i.e. next state/reward/terminality prediction with action proposal - leads to instability due to variation in both modules affecting each other and compounding.

**Audience:**

Yes

**Claims And Evidence:**

No

**Requested Changes:**

- Add more structure to intro
- Deeper analysis of when various errors come up, and what the underlying trends and mechanisms are - see "quality/weaknesses"
- Please proofread! The language is close - just needs a careful read-through to pick up on incorrect tenses and missing link words.

**Strengths And Weaknesses:**

## Clarity
### Strengths
- Methods and results section are clearly written
- Though figures should be more annotated with numbers and boundaries, but they are info-dense and straightforward.
### Weaknesses
- Intro is disorganized. Good info, but it seems like the same point about evaluating LLMs as world models for decision-making is being made over and over again. Also, lots of "however" and "on the other hand" and list-prose - difficult to follow.
- Though results are clear, since they lack depth of analysis, there is room to improve by having specific, concrete claims that could add structure to this section
- It would *really* help to have a couple example tasks and action spaces in the main text. Right now it's not immediately clear to the reader what kind of problem is actually being tested, what trajectories and proposed actions look like (is the LLM selecting from a predefined set or not?), what the state prediction looks like, etc.

## Quality
### Strengths
- Problems are well-posed. The setup makes sense for the questions the paper is asking.
- Overall, the experimentation and the paper itself are solid. The paper does what it sets out to do, and well. My concerns are with scope and depth of analysis - I don't think there's a *problem* in the paper.
- Appendix D is very useful! Multiple times in the paper I wanted to see the distribution of error rate over steps, and this gave great context.
- Policy Planning evaluation looks good
### Weaknesses
- Policy verification
  - With more steps, the gap between 4o and 4o-mini increases; accuracy overall decreases as rho increases - provide numbers, because these aren't that obvious from the charts.
  - "More steps to complete the tasks do not necessarily lead to the bad performance" - until this point, I didn't realize what finding 2 was saying. I interpret it to be about actions - having to generate a long trajectory of actions (or actions and states/reward/terminality) is harder than a short one. But having to generate state/reward/terminality for a longer sequence if actions are given, is not harder than a short one. *That makes sense*, and is what I would expect since in Policy Verification the model has ground-truth of all prior steps available. But the phrase "long-term decision-making" is vague enough and presented generally enough that at first I thought this was a contradiction. It's fair to say the first task, policy verification, is not decision-making, but warn us that that finding is not about this task.
  - For the claim that domain is more important than length for verification - these really need to be backed up with specific stats. For example, tasks where 4o has domain knowledge and 4o-mini doesn't and 4o does better, compared with tasks of the same length (or longer) where 4o-mini does have domain knowledge and does just as well. This is an example of the deeper analysis (or even just exposition) that would benefit this paper.
- Action proposal
  - Are the tasks order-fixed? Say more about them, so we know how unfair it is to compare to the ground-truth a
  - Here you say that "with the number of steps to complete the tasks, GPT-4o maintains better accuracy, while GPT-4o-mini shows a substantial drop" - this would make sense if the errors were compounding - higher error rates for later steps. But Appendix D doesn't show that, so why is this happening? Analysis here would be really interesting.
  - How much of the failure is about the diversity of the acceptable solution space not being captured by a fixed a? You mention that longer tasks have higher failure rates - I'd be curious if longer tasks are correlated with a larger set of viable trajectories, and that might be driving this.

## Originality
- Related work lacks a section on LLMs as planners. Though the setting may not be the exact same, there is extensive work using LLMs to make robust plans with various tricks that might make sense to consider - starting with [1] and going on to many more.
- Beyond that, I think the results here are useful, but the problem framework isn't that original. The paper does make a good attempt at positioning itself among prior work in terms of specifically evaluating world models, but the evaluation itself reads as similar to prior works so it seems like a distinction without a difference.

## Significance
- Especially as summarized, the findings don't seem that significant. 1 and 3 are well-known, while 2 is vague.
- That said, there does seem to be rich information in the data - it needs deeper analysis.


[1] Huang, W., Abeel, P., Pathak, D., and Mordatch, I. Language Models as Zero-Shot Planners: Extracting Actionable Knowledge for Embodied Agents. arXiv:2201.07207, 2022.

---

> ### Author Response · Authors · 2025-01-11
> **Response to xu2s**
>
> Thanks for the insightful review. Below are our responses.
>
> 1.	About the introduction. We rewrite most parts of the introduction to improve clarity, including the motivation of this paper, the three key observations motivating the three tasks, and the key results observed in the experiments. We hope this revised version of the introduction is easier to understand.
>
> 2.	About the example of the environment. Given the extreme lengths of the full prompts and the responses from GPT-4o and GPT-4o-mini, it may not be suitable for us to include them into the paper. So, we add more introduction about the environments in Section 5 and Appendix B, including the procedure to complete the tasks and the code to generate the demo actions. We note that the code to generate the demo actions include human-understandable action names. We hope these revisions can help you to understand the environments we test. We will open-source all the codes of the experiments for readers to replicate our experiments.
>
> 3.	About the action selected by the LLMs. We provide the action rules to the LLMs, e.g., turn on/off, and the LLM will use these rules and the objects provided in the state information provided in the prompt to generate the actions. Different from RL, where we need to list all actions to the agent and the agent picks one, we ask the LLMs to generate the actions by following some general rules.
>
> 4.	About the “problem” of this paper. The primary objective of this paper is to identify the importance of the world models for decision-making. The world model for decision making differs from the general world simulators or the additional modules in decision-making systems, e.g., MuZero. We believe that the world model should be the core module for decision-making. If so, how to evaluate them? Therefore, this paper presents a novel perspective to evaluate the world model, as well as the three novel tasks, i.e., policy verification, action proposal and policy planning. For clarification, we add a detailed justification of this in Appendix A.2.
>
> 5.	About the claim “More steps to complete the tasks do not necessarily lead to the bad performance”. For the traditional RL, more steps to complete the tasks, i.e., long-term decision making, usually lead to more difficult tasks, as the agent needs to explore the action spaces for learning. However, as we observed that the number of steps to complete the tasks may not be a good measure for the difficulty of the task. We revise Section 6.1 for better understanding.
>
> 6.	About the domain knowledge in the LLMs. As observed in (Wang et al., 2024), current LLMs usually have the common knowledge, but lack the scientific knowledge, where the prediction of the change of the properties after the actions is evaluated. Our findings are consistent with (Wang et al., 2024), but primarily focus on decision making.
>
> 7.	About the action proposal. As introduced in the Action Proposal paragraph in Section 4, the previous actions are added into the prompt. The LLMs need to analyze the previous actions to know the current stage of the game. With more previous actions added, the understanding of previous actions is more difficult, which may bring the accuracy drop. We revise Section 6.2 for better understanding.

---

> > ### Comment · Reviewer_xu2s · 2025-01-16
> > **Thanks for the response!**
> >
> > 1. The new sections in the intro do flow better. They are still somewhat dense and repetitive, but I don't feel as confused reading them.
> > 2. IMO most effective would be a short, representative sample of the generation, even in the appendix - much like Fig 3 in https://eureka-research.github.io/assets/eureka_paper.pdf. However, the code in the appendix is useful.
> > 3. Ah, I see. Thanks for the clarification. I'd then be curious as to how you measure match, because this makes the evaluation even more open-ended.
> > 4. I didn't take issue with your problem statement originally - I think it's worthwhile. Your distinction between "top down" for e2e gens evaluated on task, vs "bottom up" for more fundamental abilities, doesn't jive with my understanding of "top down" and "bottom up", but I do really like the focus on fundamental abilities. My main concern is that your approaches aren't specific or novel enough.
> > 5. Okay, that's what I figured and it's good to have the explanation. I'll maintain that deeper analysis would be interesting.
> > 6. Makes sense! I am inclined to believe that language model performance is heavily dependent on domain knowledge, especially wrt differences between different models (esp a pair like 4o and 4o-mini, where there are only a few other differences). My point is more that to argue that *your paper* is proving that point for the problems you test on, you'd need more careful stats.
> > 7. I see, that makes sense. I think it need more thought (and prompt strategy testing) to argue that that's a fundamental limitation, though.
> >
> >
> > Overall I think you've made some good improvements and clarifications, but the fundamental similarity to prior work still stands.

---

> ### Author Response · Authors · 2025-01-19
> **Thanks for the suggestions**
>
> Dear Reviewer xu2s:
>
> Thanks for your suggestions. Below are our responses.
>
> 1.	For point 1. Thanks. We are glad to hear that our revised version helps you to read this paper and we will continue to revise.
>
> 2.	For point 2. Thanks for the suggestions. We add two examples about the LLM’s responses about the prediction of the states and the action proposal (in Appendix C, code sample 33 and 35, highlighted as purple).
>
> 3.	For point 3. To evaluate the matching, given the actions generated by the LLM, we will use the embedding model to query the most similar action from all available actions and compare whether the targeted action is in these queried actions.
>
> 4.	For point 4 and the similarity with previous works. Thanks for the comments. World model is widely investigated in the RL literature, where the domain or game-specific world models are trained. While LLM is currently emerging as a promising method to build the general world model. Given these trend, the comprehensive evaluation about the world model with LLMs is needed. We admit that our research is inspired by the research about world model in RL (maybe that is why our work shares similarities with these previous work), and we transfer these insights into the world models with LLMs. We believe that these evaluations are important and in-time and may boost the research in this emerging field.
>
> 5.	For point 5, 6, and 7. Thanks for the comments. As a preliminary attempt, we mainly focus on the design of new evaluation tasks and the evaluations across different tasks and LLMs. For a deeper analysis, we may need to figure out the tools and measures, especially much more wide-range of tasks to test the various domain knowledge in LLMs. These analysis may be non-trivial, given the inherent sequential properties and complexities of decision making. One example is that we have the foundation model for CV and NLP, but so far, we do not have a foundation model for decision making, which is primarily due to the complexities of decision making.  We believe that our work is a step toward the comprehensive evaluation of world models, and we will continue to tackle these remaining challenges in future works.

---

### Author Response · Authors · 2025-01-11
**General Response**

We would thank all reviewers for the valuable comments and we have revised our paper accordingly. The main changes can be summarized as follows:
1. We rewrite most parts of the introduction to clarify the motivation of this work, the three key observations and the three tasks, and the key findings of our experiments. We also revise the abstract. The modified parts are highlighted as blue in the paper.

2. We slightly revise the related work.

3. We revise the introduction of the three tasks in Section 4. The modified parts are highlighted as blue in the paper.

4. We add more introduction about the environments for the evaluation in Section 5, highlighted as blue. We also add the categorization of the environments in Appendix B.1.

5. We revise the analysis of the experiment results in Section 6 to response to the comments.

6. We add Appendix A as FAQs to address the common questions related to this paper. Specifically, we add the justifications of the three tasks, the primary objective of this paper and the selection of the LLMs in this section.

---

### Author Response · Authors · 2025-01-19
**General Response 2**

Dear Reviewers and AE,

Based on the comments, we further revise the paper. The main changes are (highlighted as purple):
1. We add the examples of the generation from LLMs in Appendix C (Code sample 33 and 35).
2. We add the comparison of this work with DT and TT in Appendix A4 to avoid any confusion.

We hope this revision can further improve the paper.

---

### Author Response · Authors · 2025-01-21
**General Response 3**

Dear Reviewers and AE,

Based on the reviewer's valuable comments, we have revised the paper. The main changes are (highlighted as green):
1. We replot Figure 1 and rewrite the caption to explain the example for better understanding.
2. We rewrite some parts of the intro and the experiment sections.

We have made significant revision of the paper, where the main changes are highlighted as blue, purple and green, respectively.

Best,
Authors

---

### Decision · Action_Editor_EoSP · 2025-01-29

**Recommendation:** Reject

**Comment:**

This paper evaluates the ability of two of OpenAI’s models in predicting an environment’s next state, and reward,  as well as in choosing appropriate actions to the decision-making task. These evaluations are performed in tasks such as predicting the outcome of a sequence of actions, or composing action selection and next state / reward prediction in order to perform planning.

The topic of investigation is obviously relevant to the machine learning community and it is quite timely. Unfortunately, not so much new insight is gained from the performed evaluation. Many of the reported findings are well-known results in the literature. Additionally, another key factor in this decision is the fact that although many results were reported, the subsequent analysis was quite shallow. A thorough analysis of the results in papers such as these is crucial and, quite often, the most valuable aspect of the paper. The nuances of the problem, the failure cases, a deep understanding of the used environments, these are all things that the manuscript can bring to bear that is hard for the reader to do. Finally, the presentation itself of the manuscript still needs to be improved. The text is quite repetitive, sometimes imprecise, sentences are poorly structured, and many typos can be seen, even in the revised manuscript.

Let me expand some more on why the findings are somewhat well-known results. The superior performance of GPT-4o over GPT-4o-mini has been reported by others, including OpenAI itself (e.g., https://openai.com/index/gpt-4o-mini-advancing-cost-efficient-intelligence/). In general, the main theme of the field of LLMs so far has been that bigger models will outperform smaller ones. Thus, although the particular task in which such models were evaluated might be novel, validating a well-expected outcome is hardly a major contribution for the paper.

About the tasks themselves, as acknowledged by the authors, they are heavily inspired by the field of reinforcement learning. However, the insights of that field also make the claimed contributions somewhat limited. Specifically, the fact that prediction is not correlated with performance is well-known in the field. Results range from extremely accurate models that miss key aspects of the problem and are rendered unusable for decision-making to simple results artificially changing the predicted values in order to obtain better control (e.g., by increasing the action gap). The fact that not all predictions have the same value and that they should be maybe weighted by some distribution (likely the on-policy one) is also well-known. Examples include demonstrations of the failure modes of Go-playing systems when faced with extremely unlikely scenarios, and even the textbook definition of function approximation by Sutton & Barto (2018) where states are literally weighted differently than uniformly (and often, the used weight is the on-policy distribution) because one cannot expect to do well in every situation.

Thus, although these results are shown in the context of LLMs, they are by no means surprising because they touch upon the fundamental nature of sequential decision-making problems, which has been studied for many decades now. In fact, if the authors plan on continuing working on this topic, I recommend them to revisit more fundamental results in the field of sequential decision-making and to contextualize their results in light of such literature. Right now, the oldest reference in the paper is from 2017. Much that is relevant was done prior to 2017, but obviously, not in the context of LLMs.

To conclude referring back to TMLR’s acceptance criteria, the paper does provide evidence for its claims, but it fails to do so in a deeper way through a more careful discussion of the results. The key issue, though, is that, because the vast majority of the results of the paper are to be expected, and are just yet another confirmation of knowledge already established in the literature, this paper ends up with a quite limited audience, despite the relevance of the topic of study.

**Audience:**

I wrote a fairly detailed review comment below, so I'll refrain from writing too much here. In summary, with respect to "Audience", this paper might end up having a limited audience because the vast majority of the results of the paper are to be expected, and are just yet another confirmation of knowledge already established in the literature.

**Claims And Evidence:**

I wrote a fairly detailed review comment below, so I'll refrain from writing too much here. In summary, with respect to "Claims and Evidence", the paper does provide evidence for its claims, but it fails to do so in a deeper way through a more careful discussion of the results.